# Structural assembly of the bacterial essential interactome

**Jordi Gómez Borrego, Marc Torrent Burgas***

Systems Biology of Infection Lab, Department of Biochemistry and Molecular Biology, Biosciences Faculty, Universitat Autònoma de Barcelona, Cerdanyola del Vallès, Spain

**Abstract** The study of protein interactions in living organisms is fundamental for understanding biological processes and central metabolic pathways. Yet, our knowledge of the bacterial interactome remains limited. Here, we combined gene deletion mutant analysis with deep-learning protein folding using AlphaFold2 to predict the core bacterial essential interactome. We predicted and modeled 1402 interactions between essential proteins in bacteria and generated 146 high-accuracy models. Our analysis reveals previously unknown details about the assembly mechanisms of these complexes, highlighting the importance of specific structural features in their stability and function. Our work provides a framework for predicting the essential interactomes of bacteria and highlight the potential of deep-learning algorithms in advancing our understanding of the complex biology of living organisms. Also, the results presented here offer a promising approach to identify novel antibiotic targets.

## Editor's evaluation

This important study uses AlphaFold2 to predict the structures of bacterial protein complexes that the authors classify as "essential". The evidence supporting the conclusions is convincing, as the authors have tested the approach on an external dataset of 140 experimentally solved bacterial protein-protein complexes, 81% of which were predicted with high accuracy. This paper will be of general interest to a wide audience in the field of biosciences and in particular for molecular biologists.

*For correspondence:
marc.torrent@uab.cat

**Competing interest:** The authors declare that no competing interests exist.

## Introduction

Bacteria carry out a wide range of essential functions for their survival. These vital cellular activities are referred to as "core biological processes" and include energy production, DNA replication, transcription, translation, cell division, and cell wall synthesis, among others. These processes are executed by multiprotein complexes, which require the coordinated action of multiple essential proteins to function properly. In the absence of these proteins, the complexes cannot work, with the consequent loss of cell viability. Therefore, understanding the essential protein-protein interactions (PPIs) is critical to understand how core biological processes are regulated and how they contribute to the cell's overall function (*Koonin, 2000*; *Carro, 2018*; *Cossar et al., 2020*). By investigating these pathways and their associated proteins, we can gain insight into bacterial growth and survival mechanisms (*de Groot et al., 2020*; *Gómez Borrego and Torrent Burgas, 2022*).

Proteomic techniques such as yeast two-hybrid and tandem affinity purification coupled with mass spectrometry have identified millions of PPIs. However, the high number of false positives in high-throughput screenings makes the results less reliable (*Rao et al., 2014*; *Zhao et al., 2020*). A useful way to deal with false positives in interatomic data is to consider the three-dimensional structure of proteins, which provides insights into their function and architecture. The scientific community

has experimentally determined thousands of protein structures at atomic resolution using X-ray crystallography, NMR, and cryo-EM. However, most protein complexes have not yet been determined. Recently, novel deep-learning models such as AlphaFold2 (AF2) and RosettaFold have outperformed previous methods in predicting protein structures, providing results with similar precision to experimental methods in successful cases (*Jumper et al., 2021*; *Baek et al., 2021*). AF2 can fold protein monomers and protein complexes, outperforming standard docking approaches (*Evans et al., 2021*). Therefore, we posit that AF2 can effectively differentiate between genuine interactions and false positive cases.

The topological analysis of pathogen interactomes is a powerful method for exploring the function of interacting proteins, uncovering the evolutionary conservation of protein interactions, or identifying essential hubs (*Dong et al., 2020*; *Crua Asensio et al., 2017*; *Macho Rendón et al., 2022*). Therefore, developing a complete map of the essential interactome is a powerful strategy to study the functional organization of proteins and to identify new targets for discovering new antibiotics. Here, we used AF2 to predict the Gram-negative and Gram-positive essential interactomes, comprising a total of 1402 interactions, which include the global confidence scores of the binary complexes predicted by AF2. We also discuss how these structures can provide insight into new mechanisms of action and identify intereting PPIs to target for discovering novel antibiotics.

## Results and discussion

The average bacterial proteome is composed of ~4000–5000 proteins, which means that the interactome could potentially span around 20 million interactions. Based on recent estimates, there are approximately 12,000 physical interactions in *Escherichia coli*, which indicates that only about 0.1% of potential interactions may occur (*Rajagopala et al., 2014*). However, not all these interactions are expected to be essential for bacterial survival. If we were to selectively disrupt each interaction without impacting any other factors, only a small subset of interactions would likely be classified as essential. How can we identify these essential interactions without the paramount effort of performing all these experiments? We reasoned that a given interaction would only be essential if and only if both proteins forming the complex are essential (*Figure 1a*). While this simple approximation does not give us the exact answer, it does provide an upper bound for the essential interactome.

Using this premise, we retrieved a list of all essential Gram-negative and Gram-positive proteins from previous studies (*Figure 1b*), and considered as essential proteins only those that are present in at least two different species (*Baba et al., 2006*; *Gerdes et al., 2003*; *Goodall, 2018*; *Liberati et al., 2006*; *Gallagher et al., 2011*; *Poulsen et al., 2019*; *Ramage et al., 2017*; *Bai et al., 2021*; *Commichau et al., 2013*; *Dembek et al., 2015*; *Ji et al., 2001*; *Chaudhuri et al., 2009*; *Santiago et al., 2015*; *Liu et al., 2017*). Next, we retrieved all PPIs with experimental evidence (experimental score >0.15) and/or high-confidence PPIs (score >0.7) between these proteins from the STRING database (*Szklarczyk et al., 2021*). Additionally, we incorporated all of the synthetically lethal interactions identified in *E. coli-K12-BW25113*, as recorded in the Mslar database (*Zhu et al., 2023a*) to capture interactions between non-essential proteins that become essential in combination. We filtered out interactions that include ribosomal subunits and tRNA ligases. Using this pipeline, we modeled 722 unique Gram-negative essential PPIs (involving 216 proteins), 680 essential Gram-positive PPIs (involving 167 proteins), and 28 synthetically lethal PPIs (involving 45 proteins) using AF2-Multimer (*Evans et al., 2021*). To assess the confidence of the predictions, we used the ipTM scores to classify the models, as previously reported (*Figure 1—figure supplements 1 and 2*, *Source data 1*; *Evans et al., 2021*; *Mackay et al., 2007*; *Bryant et al., 2022a*). Concurrently, we modeled 722 Gram-negative and 680 Gram-positive negative PPIs, generated by random pairing among the selected proteins, to evaluate the ability of AF2 to distinguish between correct and incorrect models. To define an appropriate ipTM score cutoff, we calculated the cumulative distribution function of the ipTM scores for the selected and random complexes. The analysis revealed a significant difference between the two distributions (*Figure 1c*). Based on these results, we classified the models into three categories: unlikely (ipTM<0.4), plausible (0.4≤ipTM≥0.6), and high confidence (ipTM>0.6). Of the 722 Gram-negative PPIs, 549 (76.04%) were classified as unlikely, 74 (10.25%) as plausible, and 99 (13.71%) as high accuracy. For the 680 Gram-positive PPIs, 576 (84.70%) were classified as unlikely, 57 (8.48%) as plausible, and 47 (6.91%) as high accuracy (*Figure 1d*). We also validated our predicted models using crosslinking data that were available for 14 complexes (*Source data 1*). The distance restraints identified (crosslinked lysines

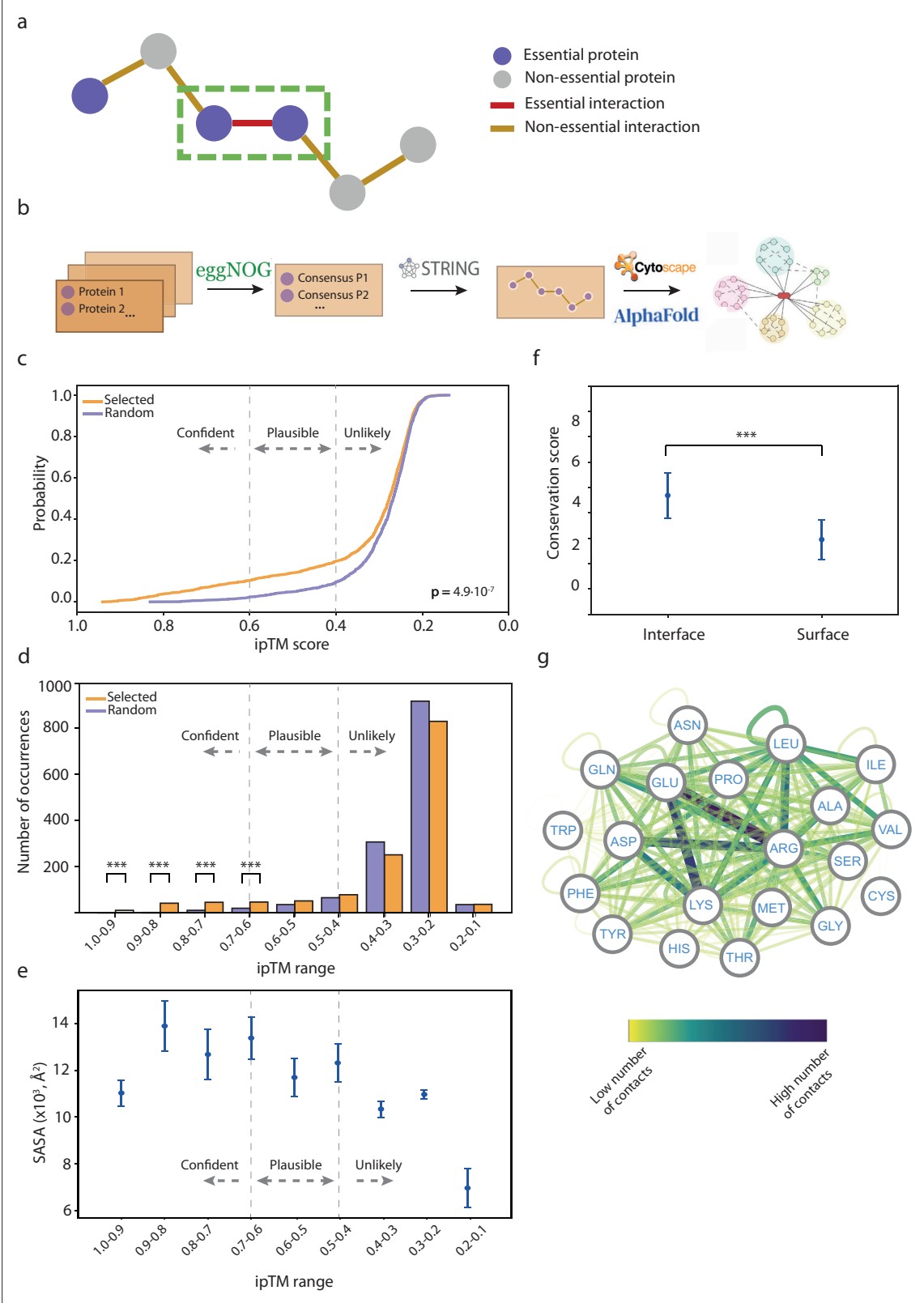

**Figure 1.** Analysis of essential binary complexes predicted by AlphaFold2 (AF2). (**a**) Representation of protein-protein interactions (PPIs) based on their essentiality. This study focuses on interactions between essential proteins, highlighted by a green rectangle. (**b**) Pipeline used to construct the essential interactomes. (**c**) Cumulative distribution function of ipTM scores in selected (orange) and randomly generated PPIs (cyan). A two-sample Kolmogorov-Smirnov test was performed to assess the statistical significance of the difference between the two distributions. (**d**) Histograms displaying ipTM

*Figure 1 continued on next page*

*Figure 1 continued*

scores in selected complexes compared to random PPIs. Chi-square test p-values: <0.05*, <0.01**, <0.001***. (**e**) Accessible surface area of AF2 binary complexes grouped by ipTM score. (**f**) Conservation score comparison between interface and surface residues. Wilcoxon test p-values: <0.05*, <0.01**, <0.001***. (**g**) Network representation of side-chain residue contacts in high-accuracy binary models. Nodes represent residue types, and edges indicate interactions between residues. The color of the edges reflects the number of occurrences.

The online version of this article includes the following figure supplement(s) for figure 1:

**Figure supplement 1.** Correlation between the ipTM score with pDockQ of high-accuracy AlphaFold2 (AF2) protein binary complexes (ipTM>0.6).

**Figure supplement 2.** Correlation between the ipTM score with pDockQ2 of high-accuracy AlphaFold2 (AF2) protein binary complexes (ipTM>0.6).

**Figure supplement 3.** AlphaFold2 (AF2) predicted interfaces colored by residue conservation.

**Figure supplement 4.** AlphaFold2 (AF2) predicted interfaces colored by residue conservation.

**Figure supplement 5.** AlphaFold2 (AF2) predicted interfaces colored by residue conservation.

**Figure supplement 6.** Venn diagram representing the number of essential proteins shared among the Gram-negative species.

**Figure supplement 7.** Venn diagram representing the number of essential proteins shared among the Gram-positive species.

are ~15–20 Å apart) are compatible with our models in 93% of the cases. Hence, despite the limited overlap between the crosslinking datasets and our list of validated interactions, for the complexes that did match, our models were consistent with the experimental data. These findings support the notion that AF2 is capable of distinguishing between incorrect and high-accuracy models, which is consistent with previous observations in other applications (*Mackay et al., 2007*). Thus, our results suggest that many of the essential PPIs retrieved from databases could be false positives, likely due to the high number of false positives found in large-scale screening experiments, which may include indirect interactions (*Zhu et al., 2023b*). We also compared ipTM scores with both pDockQ (*Akdel et al., 2022*) and pDockQ2 (*Bryant et al., 2022b*). The correlation between ipTM and pDockQ was low (R=0.328), but a stronger correlation was obtained between ipTM and pDockQ2 (R=0.649, *Figure 1— figure supplement 1*). Notably, some complexes with high ipTM values (>0.8) had minimal pDockQ2 scores, some of them virtually 0. However, these interactions showed improved pDockQ2 scores when modeled alongside accessory proteins (*Figure 1—figure supplement 2*), suggesting a better recall performance for ipTM. We conclude that pDockQ2 is a very accurate but restrictive metric. Therefore, we selected ipTM for assessing predicted interactions. Nonetheless, pDockQ and pDockQ2 scores for all predicted complexes can be found in *Source data 1*.

To test AF's predictive capabilities in bacterial complexes, we conducted a thorough validation of 140 bacterial protein-protein complexes from the PDB (*Supplementary file 1*). This dataset encompasses structures published after the latest release of AF, sharing less than 30% sequence homology with all other complexes in the PDB. According to our criteria (ipTM>0.6), we observed that 81% (113 out of 140) of these structures were accurately predicted by AF2. From all models generated, 83% (116 out of 140) were almost identical to the native structures in terms of correct folding (TM-score>0.8). Most interestingly, 72% (101 out of 140) of the predicted structures were similar in terms of root mean square deviation at the interaction interface (i-RMSD<4 Å) and 56% (79 out of 140) of the interfaces were virtually identical to the real structures (i-RMSD<2 Å), highlighting the excellent prediction power of AF2.

The interface solvent accessible surface area (SASA) of our selected models showed moderate correlation with the ipTM score, suggesting that larger interfaces were more likely to have better model accuracies (*Figure 1e*). Additionally, we considered the conservation of the interface residues, which is frequently used as a proxy to identify protein-binding sites (*Guharoy and Chakrabarti, 2010*). As expected, the residues in the interface were significantly more conserved than those located at the surface, suggesting that the predicted models are reliable (*Figure 1f*, *Figure 1—figure supplements 3–5*). We also analyzed the residue types of the interface in high-confidence models (*Figure 1g*, 4.5 Å distance cutoff). The most abundant interface residues were involved in electrostatic interactions, particularly between arginines and negatively charged residues. There was also a significant contribution of hydrophobic interactions, with a high relevance of leucine and isoleucine residues, as well as between the hydrophobic moiety of the arginine side chain and the last two residues.

In summary, we assembled a high-accuracy essential interactome for both Gram-negative (*Figure 2a*) and Gram-positive bacteria (*Figure 2b*) that will enable us to identify protein hubs and investigate the importance of these interactions. Here, we focus on new structures involving essential

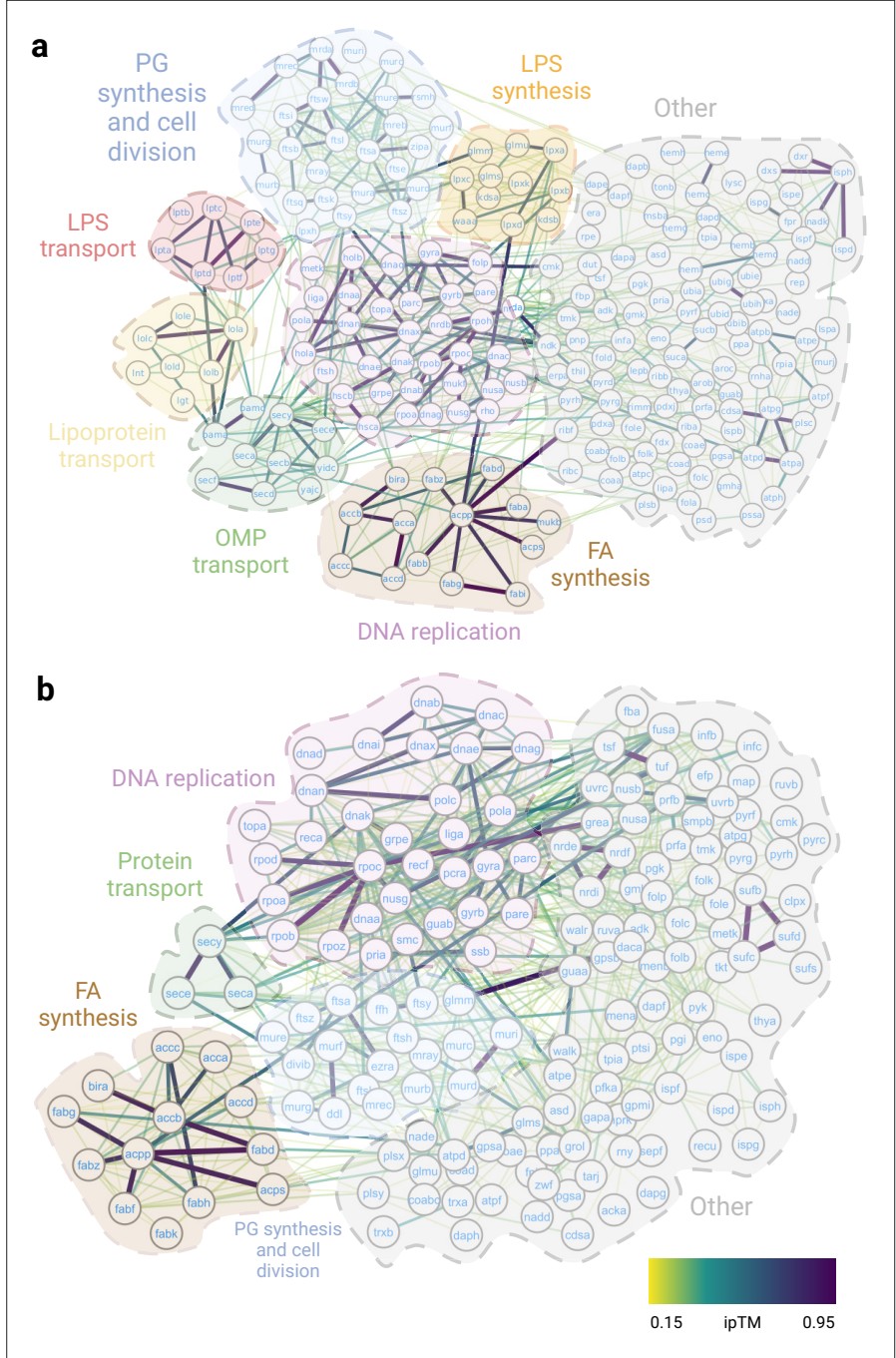

**Figure 2.** Essential interactomes. (**a**) Gram-negative essential interactome; (**b**) Gram-positive essential interactome. Nodes represent essential proteins, and edges indicate interactions between them. The color of the edges reflects the ipTM score as calculated by AlphaFold2 (AF2). The most representative biological processes are highlighted in the figure.

The online version of this article includes the following figure supplement(s) for figure 2:

**Figure supplement 1.** AlphaFold2 (AF) predicted interfaces discussed in this work aligned with experimentally solved structures.

**Table 1.** Protein complexes discussed in this work.

The ipTM score is shown along with the PDB accessions for the cases where the structure has already been solved. The AlphaFold2 (AF2) predictions are structurally aligned with the experimental structures in *Figure 2—figure supplement 1* except for SecYEDF-YidC, which is discussed in *Figure 6*.

| Protein | ipTM | PDB* | ModelArchive ID | Function |
|---|---|---|---|---|
| AccB-BirA | 0.841 | – | ma-sysbio-bei-02 | Fatty acid synthesis |
| AccABCD | 0.809 | – | ma-sysbio-bei-01 | Fatty acid synthesis |
| AcpP-FabG | 0.757 | – | ma-sysbio-bei-06 | Fatty acid synthesis |
| AcpP-FabI | 0.753 | 2FHS | ma-sysbio-bei-07 | Fatty acid synthesis |
| $AcpP_3$-$GlmU_3$ | 0.908 | – | ma-sysbio-bei-03 | Lipopolysaccharide synthesis |
| $AcpP_3$-$LpxA_3$ | 0.940 | – | ma-sysbio-bei-04 | Lipopolysaccharide synthesis |
| $AcpP3$-$LpxD_3$ | 0.957 | 4IHF | ma-sysbio-bei-05 | Lipopolysaccharide synthesis |
| LptC-LptD | 0.695 | – | ma-sysbio-bei-24 | Lipopolysaccharide transport |
| LptCAD | 0.600 | – | ma-sysbio-bei-23 | Lipopolysaccharide transport |
| SecYEDF-YidC | 0.642 | 5MG3 | ma-sysbio-bei-27 | Outer membrane protein transport |
| SecYEDFA-YidC | 0.632 | – | ma-sysbio-bei-26 | Outer membrane protein transport |
| LolA-LolC | 0.809 | 6F3Z | ma-sysbio-bei-22 | Lipoprotein transport |
| LolA-LolB | 0.838 | – | ma-sysbio-bei-21 | Lipoprotein transport |
| $FtsA_3$ | 0.761 | – | ma-sysbio-bei-13 | Cell division |
| $FtsZ_3$ | 0.614 | – | ma-sysbio-bei-18 | Cell division |
| $FtsA_3$-$FtsZ_3$ | 0.542 | – | ma-sysbio-bei-14 | Cell division |
| FtsQLBWIN | 0.727 | – | ma-sysbio-bei-17 | Cell division |
| FtsQLBK | 0.572 | – | ma-sysbio-bei-16 | Cell division |
| $FtsE_2$-$FtsX_2$ | 0.856 | – | ma-sysbio-bei-15 | Cell division |
| $MreB_4$CD-RodZ-MrdAB | 0.764 | – | ma-sysbio-bei-12 | Cell division |
| $DnaA_4$ | 0.545 | – | ma-sysbio-bei-08 | DNA replication |
| DnaN-PolA | 0.813 | – | ma-sysbio-bei-11 | DNA replication |
| DnaB-DnaI | 0.750 | – | ma-sysbio-bei-10 | DNA replication |
| DnaB-DnaC | 0.650 | 6KZA | ma-sysbio-bei-09 | DNA replication |
| NrdE-NrdF | 0.856 | – | ma-sysbio-bei-25 | DNA replication |
| GyrA-GyrB | 0.715 | 6RKU | ma-sysbio-bei-20 | DNA replication |
| GyrA-FolP | 0.847 | – | ma-sysbio-bei-19 | DNA replication |
| UbiEFGHIJK | 0.806 | – | ma-sysbio-bei-28 | Ubiquinone synthesis |

*Complexes $FtsA_3$-$FtsZ_3$ and FtsQLBK have an ipTM score <0.6 because they contain large intrinsically disordered segments that, despite not participating in the interaction, contribute to decrease the global ipTM score.

complexes, where we can gain mechanistic insight from a detailed understanding of the structure (*Table 1*).

## Complexes involved in the endogenous fatty acid synthesis

The biosynthesis of fatty acids (FA) is a crucial process for membrane biosynthesis and plays a pivotal role in related processes, such as the biosynthesis of lipid A, lipoic acid, and phosphatidic acid (*Yao and Rock, 2013*). The initial step in FA biosynthesis involves the transfer of biotin from the biotin protein ligase (BPL) BirA to the Acc complex via AccB. This is followed by the generation of malonyl-CoA

through the catalytic action of the Acc complex. The resulting malonyl-CoA is then transferred to AcpP, which couples to each step of the elongation cycle catalyzed by the Fab family of proteins, ultimately resulting in the production of FA (*Chan and Vogel, 2010*; *Figure 3a*).

Currently, the structure of the BirA-AccB binary complex remains unsolved. Hence, our model provides valuable functional insights into this complex. We show that the BPL catalytic domain of BirA aligns with the biotinyl-binding (BB) domain of AccB. Within the structure of the complex, two BirA loops play a significant role: the first loop, spanning residues 218–226, interacts with the substrate, while the second loop, consisting of residues 116–121, is enriched in arginine and aids in stabilizing the substrate's negative charge (*Figure 3b*). Based on our model, we propose that these loops act together to encapsulate the biotin moiety within the catalytic pocket of BirA, creating a closed state. Upon interaction with AccB, BirA engages with two specific AccB loops: the β-hairpin loop, that contains the important residue Lys122, and the "thumb motif", comprising residues 94–102. The presence of Lys122 near the substrate leads to electrostatic repulsion of the arginine-rich loop, creating an open state. Then, the biotin molecule can covalently attach to the Lys122 residue of AccB, presenting itself to the essential Acc complex. Our model is compatible with mutagenesis studies performed in BirA where mutations M310L and P143T were found to induce a superrepressor phenotype, i.e., BirA lacks the capacity to biotinylate AccB (*Chakravartty and Cronan, 2012*). The effect of these mutations, that do not significantly affect the BirA active site, can be explained by the destabilization of the BirA-AccB interface.

The Acc complex, composed of four subunits, is responsible for catalyzing two half-reactions. First, AccC carboxylates the biotin group attached to Lys122 of AccB. In the second step, the AccAD complex transfers the carboxyl group from Lys122-carboxybiotin to acetyl-CoA to form malonyl-CoA (*Figure 3a*). While crystal structures of all the monomeric subunits have been solved (accAD: 2F9Y, accB: 1BDO, accC: 3RV4), the full structure of the Acc complex remains unknown. The accepted stoichiometry for the Acc complex is $AccB_4C_2A_2D_2$, although a dimeric form of AccB has also been reported (*Chakravartty and Cronan, 2012*; *Cronan, 2021*). When testing various AccBC stoichiometries, we found that the dimeric form of AccB led to higher accuracies. Our predicted models suggest that the BB domain of AccB can interact with the catalytic pockets of AccA and AccC, while the N-terminal domain can only be attached to AccC (*Figure 3b*). Additionally, the essential AccB 'thumb motif' interacts with the N-terminus of AccA and the loop comprising residues 192–195 of AccC, in agreement with previous mutational and structural studies (*Tong, 2005*). These studies concluded that the thumb region is critical for identifying Acc proteins, as only biotin-dependent enzymes involved in the synthesis of malonyl-CoA contain thumb domains (*Tong, 2005*). Other studies also suggest that the thumb domain may act as a mobile lid that tightly fits into AccC and AccA active sites (*Cronan, 2001*). While the heterotetrameric AccAD has already been crystallized, we identified a new, unsolved, high-accuracy interaction between AccC and AccD, which is consistent with coevolutionary studies (*Broussard et al., 2013*). We hypothesize that this interaction is crucial for maintaining AccBC close in space with AccAD, allowing the BB domain of AccB to dynamically shuttle from AccC to AccA (*Figure 3c*). The binding affinity of the BB domain to either AccC or AccA can be influenced by the carboxylation state of the biotin moiety. The introduction of a negative charge to biotin through carboxylation may decrease the affinity for AccC, leading to the binding of the BB domain to AccA. The structural information obtained from these interfaces is consistent with the bi-substrate ping-pong mechanism followed by the Acc complex (*Cronan, 2001*).

The malonyl-CoA produced by the Acc complex is then loaded onto AcpP by FabD, initiating the FA synthesis through the catalytic reaction of FabH. The FA elongation process is cyclic and requires several Fab proteins, adding two carbons to the FA intermediate in each cycle (*Figure 3a*; *Cong et al., 2019*). The interaction of AcpP to each Fab protein is essential for the cycle to proceed, as FA intermediates are tethered and transported by AcpP (*Yao and Rock, 2015*). In these lines, many AcpP-Fab protein complexes have been solved (AcpP-FabD: 6UOJ, AcpP-FabF: 7L4E, AcpP-FabB: 6OKC, AcpP-FabA: 4KEH, AcpP-FabI: 2FHS, AcpP-FabZ: 4ZJB) but the structure of the complex AcpP-FabG remains unknown, despite the similarity between FabG and FabI (*Bartholow et al., 2021*; *Dodge et al., 2019*). Both FabG and FabI contain Rossmann folds composed of twisted β-sheets surrounded by α-helices (*Masoudi et al., 2014*). To investigate these interactions, we generated models of homodimeric FabG and FabI and analyzed their interactions with AcpP (*Figure 3—figure supplement 1*). The interfaces between the Fab homodimers exhibited a high degree of similarity,

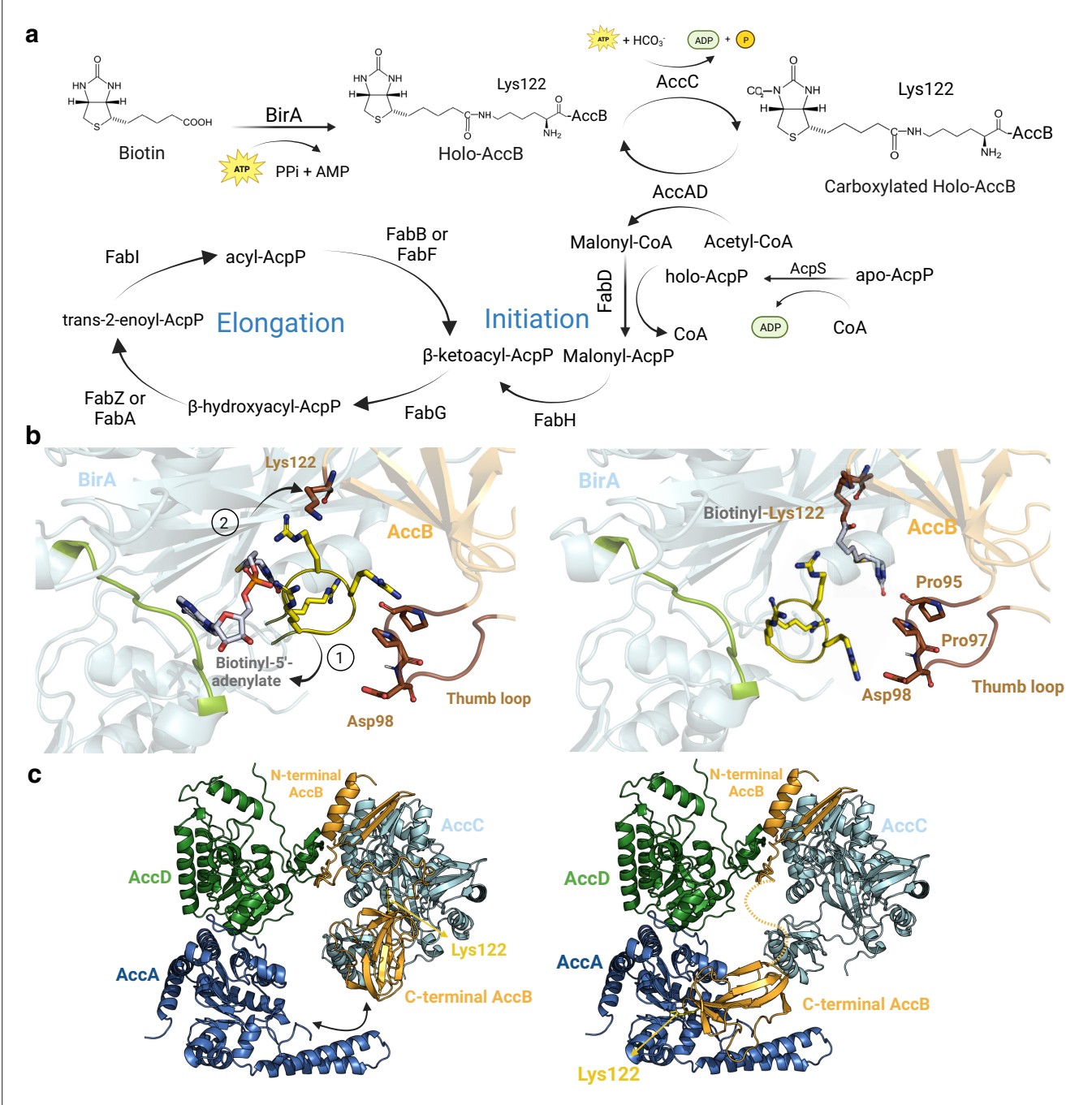

**Figure 3.** Core enzymes in fatty acid (FA) synthesis. (**a**) FA synthesis pathway. (**b**) Proposed structural rearrangements in the BirA-AccB complex. Initially, the yellow arginine-rich loop and the green loop encapsulate the substrate in BirA pocket (closed state, left). (1) Upon interaction, Lys122 in AccB repels the arginine-rich loop in BirA (open state, right), (2) facilitating the covalent binding of the substrate to Lys122. The brown thumb loop likely interacts with the arginine-rich loop, contributing to complex stabilization. (**c**) Proposed mechanism of AccB shuttle in the Acc complex. Initially, the C-terminal domain of holo-AccB exhibits stronger affinity for AccC. Once the biotinyl group of AccB is carboxylated, the same domain may shuttle to AccA, facilitating the transfer of the carboxyl group to an acetyl-CoA molecule. The dotted line represents the flexible loop of AccB that would allow it to shuttle between AccA and AccC. All represented protein structures are AlphaFold2 (AF2) models. Uniprot codes used for AF2: AccA: P0ABD5, AccB: P0ABD8, AccC: P24182, AccD: P0A9Q5, and birA: P06709.

The online version of this article includes the following figure supplement(s) for figure 3:

**Figure supplement 1.** Predicted interfaces of FabG₂-AcpP₂ (**a**) and FabI₂-AcpP₂ (**b**).

but the interaction between AcpP and the Fab partner displayed some distinct features. In both cases, Ser36 of AcpP was positioned near the active site of the FabG/FabI pocket where the catalytic activity takes place. However, the exact binding location of AcpP appeared to differ, possibly due to the presence of FabI's C-terminal region, which also interacts with the catalytic site and is absent in FabG (*Figure 3—figure supplement 1*). It is worth noting that the crystallized structure of the FabI-AcpP complex does not show AcpP's Ser36 facing the catalytic site, whereas in our model, Ser36 is positioned in the correct orientation. These findings provide valuable insights into the selectivity of AcpP for different Fab protein pairs, particularly for the uncharacterized AcpP-FabG complex.

## Complexes involved in LPS synthesis

Lipopolysaccharide (LPS) is a crucial molecule that forms the outer leaflet of the Gram-negative outer membrane (OM). It consists of lipid A, O-antigen polysaccharide, and a core oligosaccharide connecting both parts. The OM is an asymmetric lipid bilayer, with LPS making up the outer leaflet and phospholipids forming the inner leaflet. The biosynthesis of lipid A, also called the Raetz pathway, is highly conserved in Gram-negative bacteria and involves several enzymes of the Lpx family (*Shanbhag, 2019*; *Whitfield and Trent, 2014*). In *E. coli*, LpxA binds to AcpP to transfer β-hydroxymyristoyl, one of the many substrates of FabA/FabZ, to UDP-*N*-acetylglucosamine, which is synthesized by GlmU. Next, LpxC deacetylates the LpxA product, and LpxD transfers another β-hydroxylauroyl molecule, which is also transported by AcpP. The Raetz pathway requires six more reactions to convert the initial UDP-*N*-acetylglucosamine into Kdo2-lipid A before it is translocated to the outer leaflet of the inner membrane (IM) by the MsbA flippase (*Figure 4a*; *Shanbhag, 2019*; *Mahalakshmi et al., 2014*).

The crystal structures of homotrimeric $LpxA_3$ (6P9S), $LpxD_3$ (6P89), and $GlmU_3$ (2OI6) contain left-handed β-helix domains, with different structural features characterizing each protein (*Figure 4b*). Though the $LpxD_3$-$AcpP_3$ structure is already known (4IHF), the $LpxA_3$-$AcpP_3$ and $GlmU_3$-$AcpP_3$ complexes remain unsolved. The interfaces in our predicted models for both complexes consistently display the critical Ser36 residue of AcpP (located in the universal recognition helix or helix II) placed in the catalytic chamber, resembling the $LpxD_3$-$AcpP_3$ crystal structure. Interestingly, our models reveal a hydrophobic patch that accommodates the lipid moiety of the ligand (*Figure 4—figure supplement 1*) with a size proportional to the substrate's length. These structures reveal that all the complexes contain an extruding loop derived from the left-handed β-helix domain, which could act as a lid, facilitating ligand recognition. Therefore, we propose that a shared mechanism mediated by the extruding loop of the left-handed β-helix domain defines substrate specificity in these three complexes.

## Complexes involved in LPS transport

The lipid A-core synthesis and transport in bacteria must be tightly coupled. The lipid A-core region of LPS is synthesized in the cytoplasm and transported to the periplasmic face of the IM using the MsbA flippase. The O-antigen is then ligated to the lipid A-core by the WaaL ligase to form the LPS molecule. Subsequently, the LPS is carried from the IM to the OM by the lipoprotein transport protein complex (LptA-G), which plays a vital role in cellular function (*Olsen et al., 2007*; *Hicks and Jia, 2018*; *Putker et al., 2015*).

To extract the LPS from the IM, the $LptB_2FG$ complex, an ATP-binding cassette (ABC) transporter, hydrolyzes ATP to induce conformational changes in the transmembrane (TM) LptFG complex. The LptFG periplasmic β-jellyroll (βJR) domains are arranged in an antiparallel manner, creating a conduit for the LPS to move from the hydrophobic pocket of LptFG to the βJR domains of LptFC (*Figure 5a*). Once inside the LptFC complex, LptA facilitates the unidirectional transport of LPS to LptD in the OM. For this transport the formation of a physical bridge in the periplasm between LptC, LptA, and LptD is essential (*Li et al., 2019*). Hence, LPS undergoes a two-portal mechanism, moving from LptA to the N-terminal βJR fold of LptD, and then to the C-terminal TM β-barrel domain. There, the LptDE complex forms a plug-and-barrel structure, with LptE inserted into the β-barrel of LptD, effectively blocking a portion of the extracellular opening to maintain membrane impermeability (*Figure 5a*; *Okuda et al., 2016*).

While the cryo-EM and crystal structures of LptA (6GD5), $LptB_2FGC$ (6MK7), and LptDE (4RHB) have been extensively studied, the structure of the bridge formed by LptCAD remains ill-defined. Additionally, the exact number of LptA molecules that make up the periplasmic bridge is still unknown, although previous research suggests that LptA molecules in isolation can form polymers of up to eight

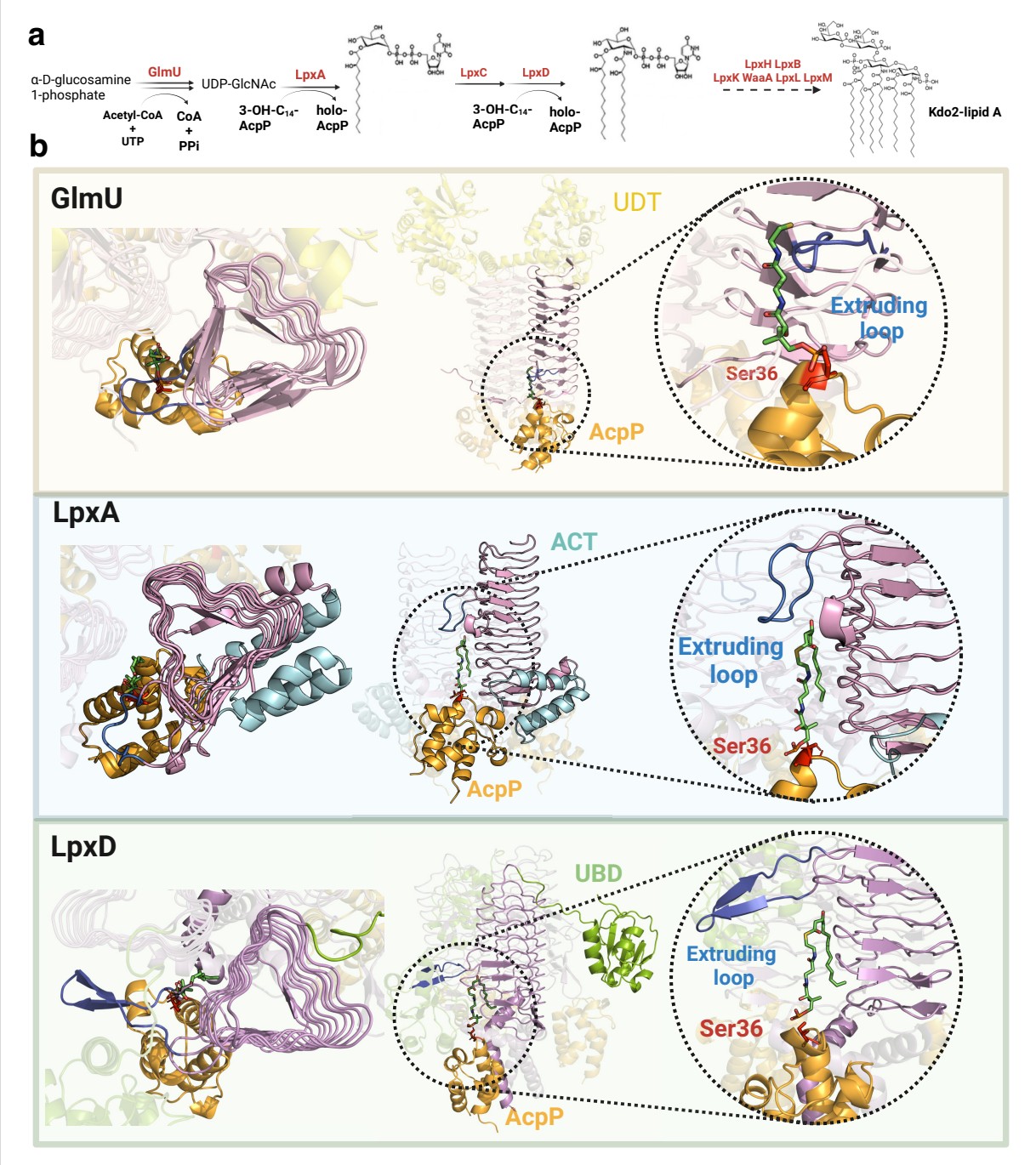

**Figure 4.** Common mechanism in initial steps of lipopolysaccharide (LPS) synthesis pathway. (**a**) Simplified Raetz pathway. (**b**) Top view (left), front view (center), and magnified interface (right) of GlmU-AcpP, LpxA-AcpP, and LpxD-AcpP predicted AlphaFold2 (AF2) models. GlmU contains an N-terminal uridyltransferase domain (UDT, yellow) while LpxA incorporates a C-terminal acetyltransferase domain (ACT, cyan) forming a collapsed helix that does not interact with the other LpxA monomers. LpxD incorporates a uridine-binding domain (UBD, green) and a C-terminal acetyltransferase domain forming a 3-helix bundle. The common left-handed β-helix domain is colored in pink, the extruding loop is highlighted in blue, AcpP in orange, and AcpP's Ser36 in red. Uniprot codes used for AF2: GlmU: P0ACC7, LpxA: P0A722, LpxD: P21645, AcpP: P0A6A8.

The online version of this article includes the following figure supplement(s) for figure 4:

**Figure supplement 1.** Electrostatic potentials of AlphaFold2 (AF2) predicted models for the GlmU-AcpP (**a**), LpxA-AcpP, (**b**), and LpxD-AcpP (**c**) complexes.

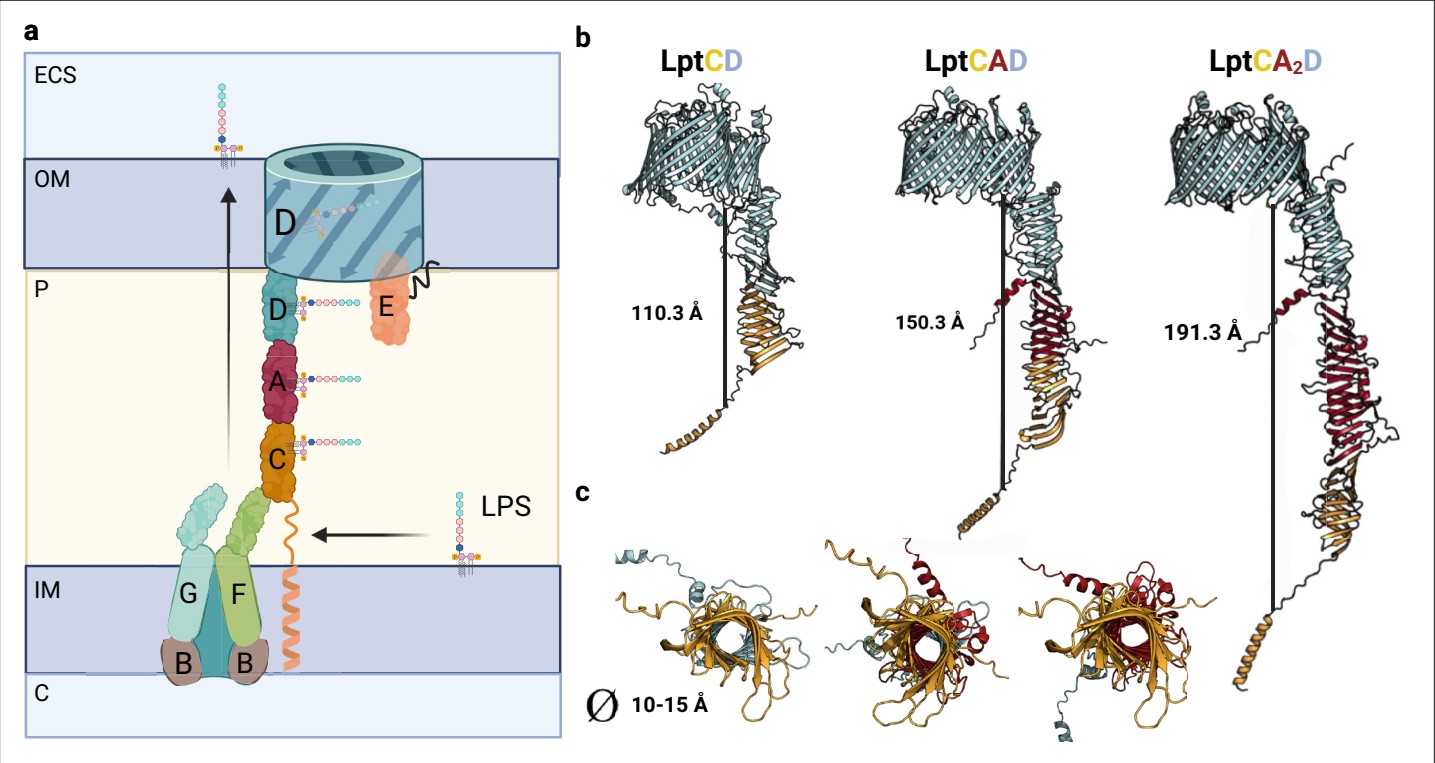

**Figure 5.** Model of Lpt bridge. (**a**) Schematic representation of the Lpt complex. Initially, the LptB$_2$FGC complex extracts the LPS from the inner membrane (IM). The LPS molecule then moves from the hydrophobic pocket of LptFG to LptC. The LptCAD periplasmic bridge shields the LPS molecule and facilitates its insertion into the outer membrane (OM) by LptDE. Key compartments include the IM, OM, periplasm (P), cytoplasm (C), and extracellular space (ECS). LPS refers to lipopolysaccharide. (**b**) AlphaFold2 (AF2) models of Lpt bridges with varying LptA stoichiometries are depicted, with each LptA subunit approximately measuring 40 Å in length. (**c**) A view of the interior of the Lpt bridge reveals a hole with a diameter ranging from 10 to 15 Å in all three cases. The structures are presented in the same order as in the previous model: LptCD, LptCAD, and LptCA$_2$D. Uniprot codes used for AF2: LptA: P0ADV1, LptC: P0ADV9, LptD: P31554.

subunits (*Malojčić et al., 2014*; *Merten et al., 2012*). In our study, we have successfully generated a high-accuracy model of the periplasmic bridge by computationally predicting the structure of the LptCAD complex. Our model supports the formation of a head-to-tail LptCAD complex (*Figure 5b*) and suggests that the presence of a single LptA monomer is enough to form a bridge spanning approximately 15 nm, which corresponds to the average thickness of the periplasm in *E. coli*. It should be noted that the width of the periplasmic space can vary depending on environmental conditions, contracting or expanding during stress (*Santambrogio et al., 2013*). Consequently, the oligomeric state of LptA may adapt to these changes, allowing the formation of larger bridges. By modeling different LptA oligomers (such as LptA$_2$ and LptA$_3$), we were able to generate models consistent with previously reported structures (*Merten et al., 2012*; *Sochacki et al., 2011*), indicating that LptA can transiently oligomerize in the periplasm, facilitating the formation of extended bridges (*Figure 5b*). Furthermore, under certain conditions, the periplasmic space can significantly shrink (approximately 10 nm), consistently with the loss of a single LptA molecule. In our analysis, we identified a high-accuracy interaction between LptC and LptD, which involves the interface region of their βJR domains, analogous to previously characterized complexes, suggesting that the formation of the complex without LptA is also feasible.

## Complexes involved in OMP transport

Outer membrane proteins (OMPs) are β-barrel proteins that are synthesized in the cytoplasm and require translocases to be transported to the OM (*Suits et al., 2008*). This transport is mediated by the Sec complex, which drives the translocation of unfolded peptides across the IM, and the Bam machinery, which mediates the insertion and folding of β-barrel proteins into the OM. High-resolution

cryo-EM images of the Bam complex are available, but only a single low-resolution (5MG3, 14 Å) structure of the Sec holo-translocon (HTL; SecYEGDF-YidC).

The export of nascent OMPs can occur co-translationally if the proteins contain signal peptides or post-translationally through the action of SecA (*Knyazev et al., 2018*). The translocation process relies on the essential components SecY and SecE. While SecY and SecE are essential for translocation, SecG stimulates the process but is not indispensable. SecY and SecE interact with other accessory proteins such as SecDF, a secretion factor that utilizes proton motive force to facilitate protein secretion into the periplasm, and YidC, an integral membrane protein that functions as a chaperone and insertase for membrane protein biogenesis (*Figure 6a*; *Ma et al., 2019*). Crystal and cryo-EM data have provided valuable insights into the structure and function of sub-complexes like SecYEA (6ITC), SecYEG (6R7L), SecDF (3AQ0), and YidC (6AL2), but limited information is available regarding the conformational rearrangements carried out by YidC within the overall structure of the translocon (*Suits et al., 2008*; *du Plessis et al., 2011*; *Oswald et al., 2021*).

To gain a more comprehensive understanding of the translocon assembly, we generated a model of the HTL assembly, which encompasses SecYEDF and YidC, and compared it to the low-resolution cryo-EM structure (*Figure 6b*; *Veenendaal et al., 2004*). Interestingly, the model positioned the previously uncharacterized N-terminal helix of YidC inside the central cavity, providing potential stabilization of the complex in a specific state (*Figure 6—figure supplement 1*). In the cryo-EM structure, the C-terminal domain of SecE encircles SecY from the external face (*Figure 6b*, top). However, in the model, SecE adopts a diagonal embrace of the two SecY halves, with the hinge facing the central cavity and the C-terminal region facing the TM domains of YidC (*Figure 6—figure supplement 1*). The cryo-EM structure shows close contacts between SecF and YidC, constraining the complex and preventing the formation of the central cavity. In contrast, our model shows weak interaction between SecF and YidC's N-terminal helix. In addition, SecF is distant from the TM and periplasmic domains, being SecD positioned between both subunits. Furthermore, the crystal structures of SecDF and YidC closely resemble our model but exhibit poor alignment with the cryo-EM structure (RMSDs for YidC and SecDF: 0.512 Å and 3.552 Å in our model; 14.060 Å and 15.336 Å in the cryo-EM structure).

The subunit organization in our model is consistent with a proposed mechanism in which the preprotein infiltrates into the pocket of SecY, displaces the plug domain, and is subsequently released through the exit lateral gate, with the dynamic periplasmic domains coordinating its release into the periplasm. Previous studies have examined the dynamics of the SecY lateral gate (formed by TM2 and TM7) and concluded that it fluctuates significantly, irrespective of the bound ligand and the experimental conditions (*du Plessis et al., 2011*). In the cryo-EM structure, the lateral gate is in a closed state and faces the membrane, whereas in our model, it faces the TM region of YidC (*Figure 6b*).

We also decided to model the HTL including SecA as several mechanisms have been proposed to explain post-translational translocation in bacteria (*Figure 6—figure supplement 1*; *Knyazev et al., 2018*). Tight interactions involving the SecY's β-hairpin loop comprising residues 247–262 and SecA could explain some rearrangements in SecY that mediate the open/closed states, allowing the preprotein to move from the SecA-SecY pocket to the SecY pore. It is noteworthy that when SecA attaches to SecY, the central cavity is not formed, and the N-terminal helix of YidC is positioned near the lateral exit gate of SecY, which supports earlier research (*Figure 6—figure supplement 1*; *Botte et al., 2016*). It appears that the arrangement of the Sec translocon can vary greatly and depends on its interaction with SecA, and the ribosome, and whether the translocation is YidC-dependent or -independent. Based on our models, SecA is essential for propelling the polypeptide during the initial stages, and the preprotein is transported to the exit lateral gate where YidC is located. If SecA is absent, a different mechanism may be employed to translocate the preprotein, (*Knyazev et al., 2018*; *Steudle et al., 2021*; *Alvira et al., 2020*) and the N-terminal helix of YidC found in the central cavity may play a crucial role.

## Complexes involved in lipoprotein transport

Lipoproteins are integral components of the OM that play essential roles in cell wall synthesis, secretion systems, and antibiotic efflux pumps (*Tsirigotaki et al., 2017*). The transport of lipoproteins from the IM to the OM is facilitated by the Lol pathway, which involves five essential proteins: LolA, LolB, LolC, LolD, and LolE (*Figure 7a*; *Grabowicz and Silhavy, 2017*). However, recent studies suggest that

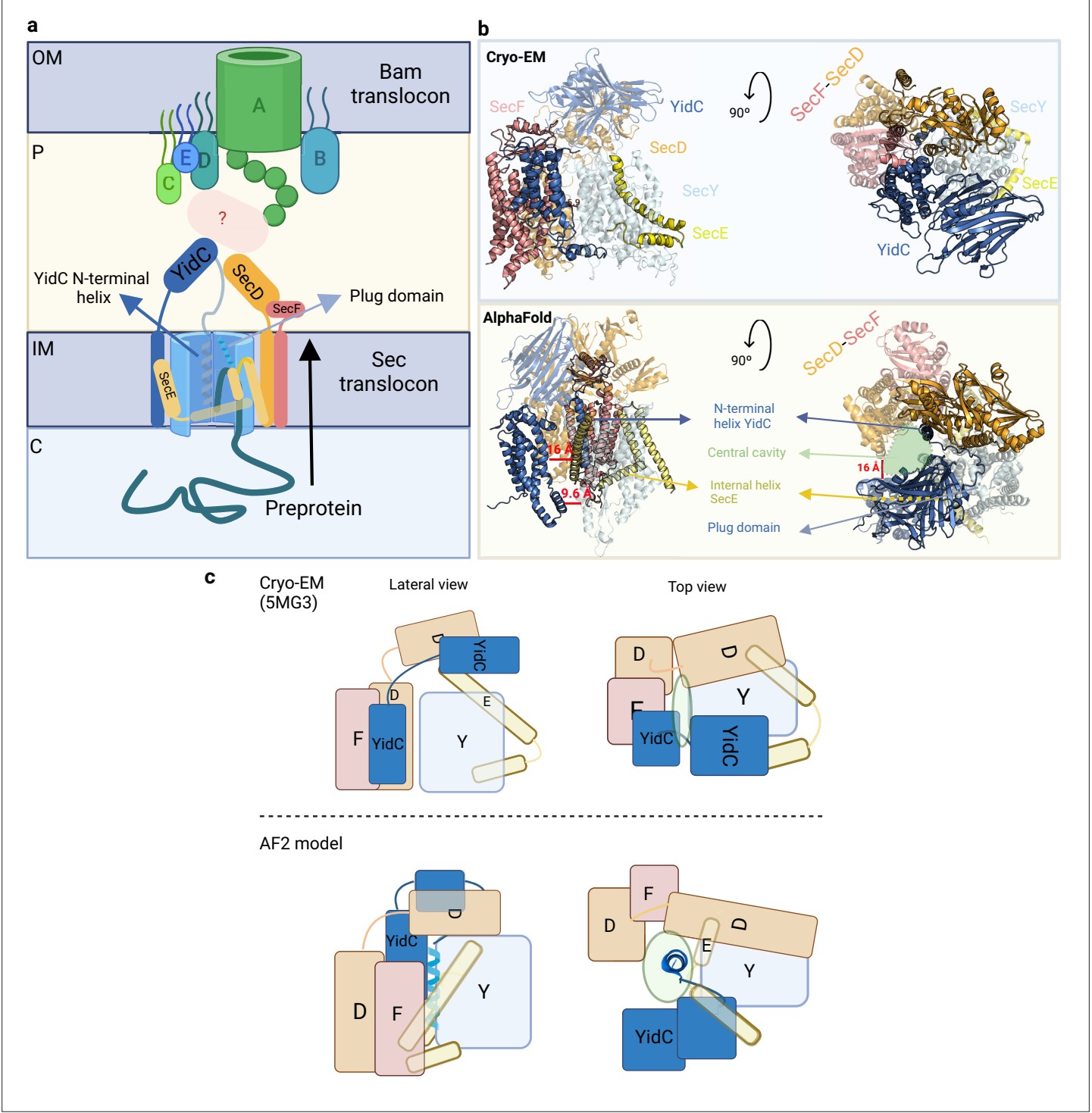

**Figure 6.** Organization of the Sec translocon. (**a**) Schematic representation of the Sec translocon and its crosstalk with the Bam translocon. During protein translocation, the preprotein engages with the central cavity of SecY, where the N-terminal helix of YidC is accommodated. Subsequently, the plug domain is displaced, allowing the preprotein to be released into the periplasm through the lateral gate. Crosstalk between the Sec and Bam translocons may occur via indirect interactions facilitated by periplasmic chaperones. Key compartments include the inner membrane (IM), outer membrane (OM), periplasm (P), and cytoplasm (C). (**b**) Front and top views of the cryo-EM structure (top) and the AlphaFold2 (AF2) model (bottom), providing different perspectives on the Sec translocon organization. (**c**) Schematic representation of the Sec translocon showing the relative orientation of the corresponding subunits in the cryo-EM structure (top) and our AF2 model (bottom). Uniprot codes used for AF2: secD: P0AG90, secE: P0AG96, secF: P0AG93, secY: P0AGA2, YidC; P25714.

*Figure 6 continued on next page*

*Figure 6 continued*

The online version of this article includes the following figure supplement(s) for figure 6:

**Figure supplement 1.** Sec translocon bound to SecA.

in certain species, the involvement of LolA and LolB in lipoprotein trafficking may not be essential, indicating the existence of alternative pathways (*Tsirigotaki et al., 2017*).

In the Lol pathway, lipoproteins are extracted from the IM by the ABC transporter $LolCD_2E$ and transferred to the lipoprotein periplasmic carrier, LolA. The ATPase activity of the LolD dimer is

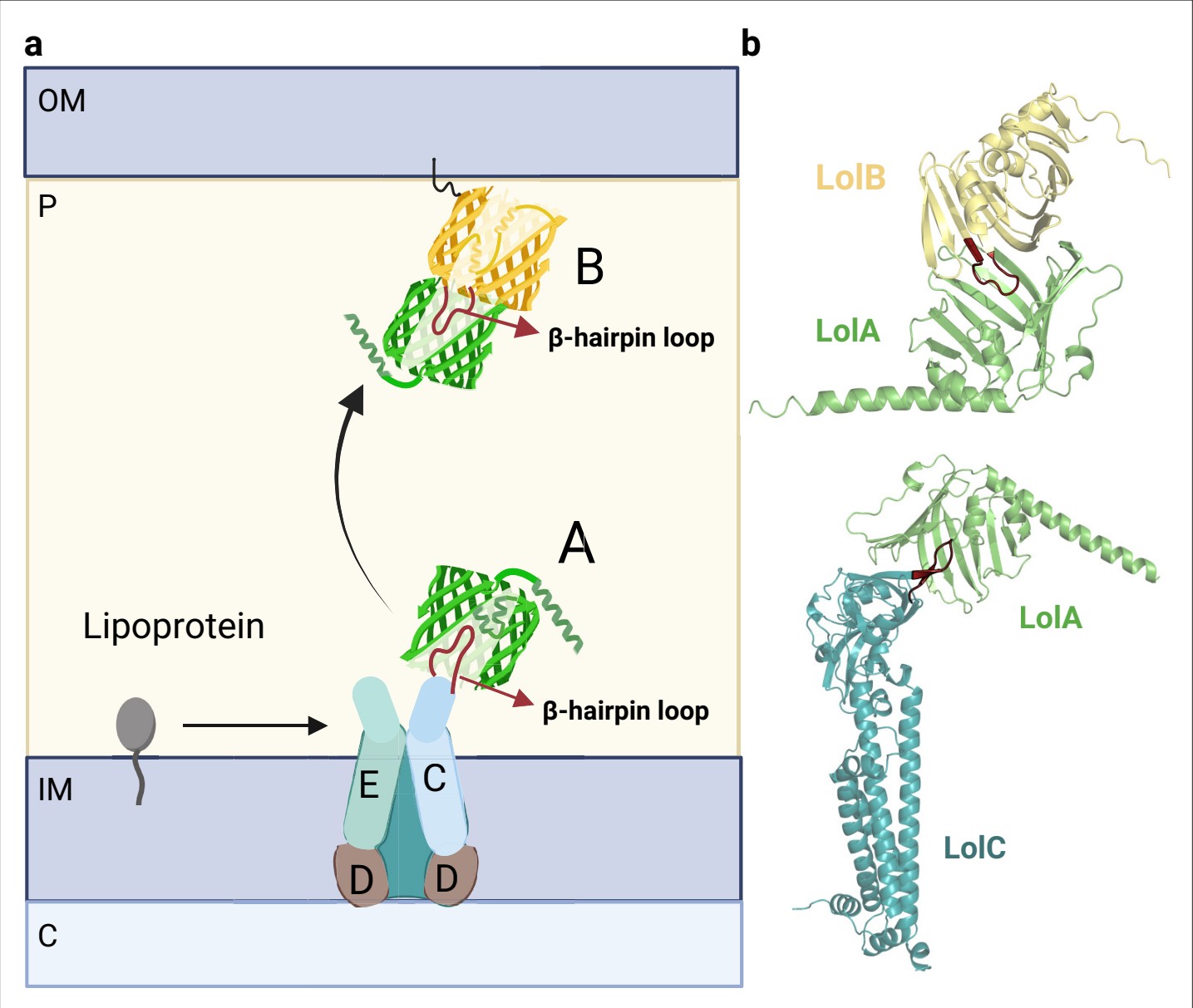

**Figure 7.** Organization of the Lol complex. (**a**) Schematic depiction of the Lol complex. The outer membrane (OM), inner membrane (IM), periplasm (P), and cytoplasm (C) are highlighted in the figure. The structures of LolA and LolB are shown in green and yellow, respectively. The $LolCD_2E$ complex and the lipoprotein are represented in a schematic manner. (**b**) Predicted AF2 models of LolAB and LolAC. The protruding loops of LolB and LolC are highlighted in red for clarity. Uniprot codes used for AF2: lolA: P61316, lolB: P61320, lolC: P0ADC3.

The online version of this article includes the following figure supplement(s) for figure 7:

**Figure supplement 1.** Predicted interfaces of LolA with LolC and LolE.

responsible for ATP hydrolysis, leading to structural rearrangements that enable LolC to recruit LolA (*Figure 7b*, bottom) (*Narita and Tokuda, 2017*). LolA then accepts the lipoprotein moiety. Despite sharing structural homology, LolC and LolE have two distinct clear roles: LolC specifically binds to LolA, while LolE interacts with lipoproteins (*Kaplan et al., 2018*) To gain insights into the specific role of each subunit, we compared the already solved LolAC structure (6F3Z) with the hypothetical LolAE complex (*Figure 7—figure supplement 1*). LolC and LolE share an identical overall fold, except for a β-hairpin located in the interface. The β-hairpin loop in LolC is smaller and can be easily accommodated within the β-barrel of LolA. Instead, the loop in LolE is larger and cannot be placed inside the β-barrel. This comparison indicates that the β-hairpin loop may be responsible for the specific interaction between LolA and LolC.

After the lipoprotein is loaded into LolA, the lipoprotein-LolA complex travels across the periplasm to interact with LolB, which accepts the lipoprotein and incorporates it into the OM. LolA and LolB also contain a β-barrel domain, however, the latter also accommodates a helix inside the β-barrel (*Kaplan et al., 2022*). Surprisingly, the LolAB crystal structure remains unsolved. Our LolAB model shows strikingly similar interfaces with LolAC, as both show the protruding β-hairpin loop contained inside the β-barrel hydrophobic cavity, evidencing that both complexes share a similar mechanism (*Figure 7b*, top). Moreover, the critical Leu68 of LolB, which is crucial to receive and localize lipoproteins to the OM, is located at the interface region (*Takeda et al., 2003*). An incorrect fold is obtained if one tries to model the interaction between LolB and LolC (*Figure 7—figure supplement 1*) as the protruding β-hairpin loops of both subunits face each other instead of following a 'mouth-to-mouth' model. Probably the helix inside the LolB β-barrel allows LolC to distinguish between LolA and LolB as binding partners. In summary, this data is consistent with a model in which the periplasmic chaperone LolA accepts and delivers lipoproteins in a 'mouth-to-mouth' mechanism by interacting specifically with LolC and LolB (*Narita and Tokuda, 2017*).

## Complexes involved in cell division

Bacterial cell division is a highly regulated and dynamic process that involves the coordinated action of numerous proteins. The initial step of this process is the formation of the Z-ring, a circular structure located at the midcell, composed of polymerized tubulin-like FtsZ proteins, which serves as a landmark for the division site. FtsA and ZipA proteins anchor the FtsZ proteins to the membrane (*Hayashi et al., 2014*). Current models suggest that other proteins like FtsN, FtsK, and the FtsQLB complex are recruited when FtsA changes from a group to a single molecule through FtsEX (*Hayashi et al., 2014*; *Mahone and Goley, 2020*). These recruited proteins are important for initiating the contraction of the membrane. Later, FtsN recruits FtsW, which adds glycan strands, and FtsI, which connects peptide side chains to specific areas where peptidoglycan (PG) is needed (*Figure 8*). FtsW and FtsI contribute to the synthesis and modification of the cell wall during cell division (*Pichoff et al., 2019*; *Rohs et al., 2018*).

The crystal structure of FtsA bound to the C-terminal helix of FtsZ of *Thermotoga maritima* is already solved (4A2A) but the N-terminal GTPase domain and the long-unfolded linker which connects both domains of FtsZ in the complex are missing. AF2 allowed us to predict the FtsA-FtsZ binary complex including the interface region between the GTPase domain of FtsZ and FtsA, absent in the crystal structure. After testing multiple stoichiometries, we detected that trimeric and tetrameric FtsA and FtsZ are the most confident states based on the ipTM score. The FtsA$_4$-FtsZ$_3$ complex displays the C-terminal of FtsZ attached to the pockets created between two FtsA monomers (*Figure 8*).

Although FtsZ plays a central role in cell division, the divisome assembly depends on the recruitment of multiple scaffold proteins and is influenced by the polymerization states of FtsA and FtsZ. Furthermore, some essential proteins like FtsN and FtsX were not included in our essential interactome as they were identified as essential in only one species, *E. coli*. With the aim of increase our understanding of the cell division process, we decided to include these proteins in our model. Also, we successfully obtained a high-confidence model for the experimentally unsolved FtsEX complex, an ABC transport involved in coordinating PG synthesis and hydrolysis and recruiting divisome proteins (*Figure 8—figure supplement 1*; *Mahone and Goley, 2020*). Recent studies have suggested that FtsEX acts on FtsA, promoting the transition from polymeric to monomeric FtsA, which in turn activates the constriction pathway through its interaction with FtsN (*Hayashi et al., 2014*; *Mahone and Goley, 2020*). Unfortunately, our attempts to predict the interfaces between FtsEX and FtsA/FtsZ

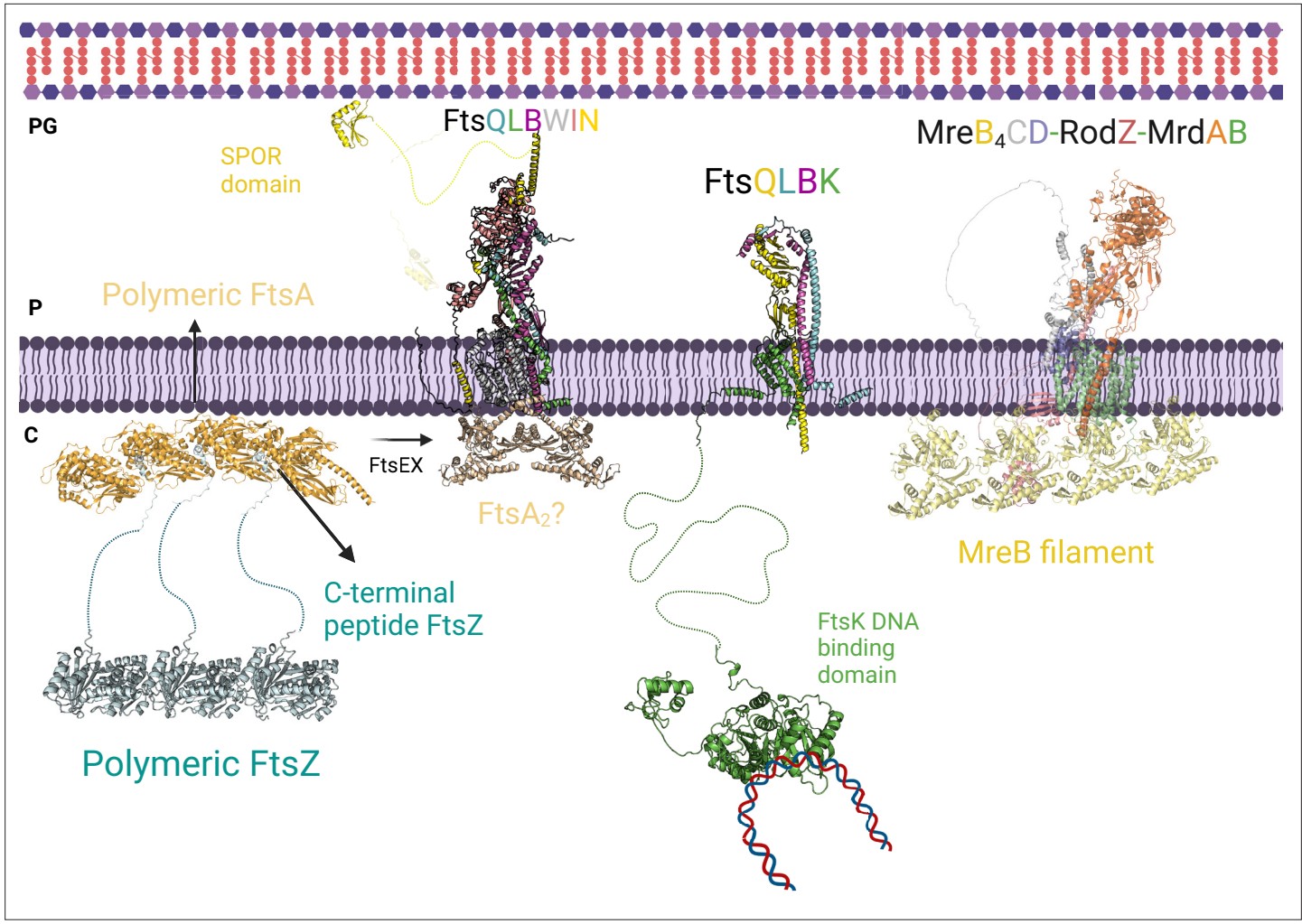

**Figure 8.** Divisome and elongasome predicted complexes. The initial step of cell division involves the binding of the polymer FtsZ to inner membrane proteins FtsA. FtsEX assists in converting the polymer form of FtsA to its individual subunit form, which promotes the recruitment of FtsK, FtsQLB, FtsWI, and FtsN. On the left side, the AlphaFold2 (AF2) model shows the interaction between FtsQLBWIN and FtsA$_2$. Previous research suggested that the monomeric form of FtsA is responsible for recruiting the divisome proteins, while the AF2 model indicates that the dimeric form of FtsA could also play a role in this recruitment. In the center, the interactions between the transmembrane domains of FtsK and FtsQLB are shown, along with FtsK's long linker and the DNA binding domain. This interaction likely occurs before the recruitment of FtsN to prevent DNA entrapment during division. On the right side, the AF2 predicted elongasome complex is displayed. For a more detailed depiction of the divisome and elongasome complexes, please refer to *Figure 8—figure supplement 2* and *Figure 8—figure supplement 3*, respectively. Notations: PG refers to peptidoglycan, P refers to periplasm, and C refers to cytoplasm. All represented protein structures are AF2 predictions. Uniprot codes used for AF2: ftsA: Q02KT7, ftsB: A0A0H2ZE93, ftsE: A0A0H2ZGN1, ftsH: A0A0H2ZC79, ftsI: A0A0H2ZFM0, ftsK: P46889, ftsQ: A0A0H2ZGP2, ftsN: P29131, ftsW: A0A0H2ZGG8, ftsY: A0A0H2ZKT5, ftsZ: A0A0H2ZM25. mrdA: P0AD65, mrdB: P0ABG7, mreB: P0A9X4, mreC: P16926, mreD: P0ABH4, rodZ: P27434.

The online version of this article includes the following figure supplement(s) for figure 8:

**Figure supplement 1.** AlphaFold2 (AF2) model of the FtsE$_2$X$_2$ complex.

**Figure supplement 2.** Detailed view of AlphaFold2 (AF2) divisome model.

**Figure supplement 3.** Detailed view of AlphaFold2 (AF2) elongasome model.

were unsuccessful. We also modeled the binary complexes, FtsQB and FtsBL, which strongly support the formation of the FtsQLB complex. FtsLB adopts a helical coiled-coil conformation, while FtsQB reveals the binding of FtsB's C-terminal domain to FtsQ, consistently with other experimental findings (*Figure 8—figure supplement 2*; *Vicente et al., 2006*). Additionally, we explored the interactions between FtsK and FtsQLB and found that their binding is primarily mediated by the N-terminal TM domains of FtsK and FtsQ (*Figure 8*). We observed contacts between the C-terminal domain of FtsK and the periplasmic domains of FtsQLB. These findings suggest that FtsKQ could play a

role in connecting chromosome segregation and PG synthesis, ensuring DNA is not trapped during membrane constriction.

Our interactome highlights the central role of FtsW, which participates in multiple PPIs. As previously mentioned, FtsW and FtsI form a well-studied GTase-TPase pair involved in PG synthesis (*Pichoff et al., 2019*; *Craven et al., 2022*). The current model of cell membrane constriction proposes that FtsQLB mediates the localization of FtsWI to the midcell and triggers the final steps of constriction, although its structure remains structurally unverified (*Vicente et al., 2006*). We obtained confident models when modeling FtsW with FtsL and FtsB, which are consistent with a model in which the formation of FtsQLB regulates FtsWI, as detailed in recent studies (*Vicente et al., 2006*). Finally, FtsN is an essential protein involved in initiating membrane constriction through interactions with FtsQLB and FtsWI sub-complexes (*Hayashi et al., 2014*). Therefore, we extended our analysis to predict the structures of the FtsWIN and FtsQLBWIN complexes. As shown in *Figure 8—figure supplement 2*, the N-terminal helix of FtsN interacts with the TM helices of FtsW, while the helix and loop comprising residues 98–140 attach to the C-terminal domain of FtsI. The SPOR domain of FtsN does not participate in protein interactions. In addition, we acquired an FtsQLBN model with poor precision, suggesting that FtsN would bind exclusively to FtsWI. Notably, we observed that the SPOR domain of FtsN (present in the FtsWIN model) shares the same interaction site as FtsLB when joining with FtsWI (as seen in the FtsQLBWI model) by overlapping the FtsWIN and FtsQLBWI structures. Therefore, we suggest that PG synthesis occurs when FtsQLB binds to FtsWI, displacing the SPOR domain so that it can attach to PG, facilitating the transport of the complex to regions where PG is required.

## Complexes involved in cell elongation

The elongasome is formed when the actin-like MreB protein polymerizes and attracts various proteins from the Mre and Rod families, which are critical for maintaining the shape of rod-shaped bacteria, such as *E. coli* (*Hayashi et al., 2014*; *Sjodt et al., 2020*). In these bacteria, the elongation and cell division are closely coordinated, to avoid changes in shape that may impact cell survival (*van Teeseling, 2021*). The elongasome and divisome share important similarities: both involve the polymerization of an actin-like protein that signals the assembly of membrane-associated protein complexes anchored in the IM, such as FtsA and MreB (*van Teeseling, 2021*). These proteins form dynamic filaments with an actin-like nucleotide-binding domain that hydrolyzes ATP to initiate polymerization (*van Teeseling, 2021*). Both complexes also have specific GTase-TPase sub-complexes which polymerize and cross-like glycan chains: FtsWI in the divisome and MrdAB in the elongasome. However, while MrdAB is mainly found in the lateral wall and midcell, FtsWI is localized in the division septum (*Szwedziak and Löwe, 2013*). Despite their similarities, the structure of the two complexes differs in several ways. The divisome comprises the tubulin-like FtsZ protein which assembles in a ring-like complex and recruits several Fts proteins such as FtsWI, FtsEX, FtsQLB, FtsK, and FtsN (*Hayashi et al., 2014*). In contrast, the elongasome contains the actin-like MreB-forming patches attached to the membrane and interacts with proteins such as RodZ, MreBCD, and MrdAB (*Graham et al., 2021*). Moreover, while MreB is undoubtedly an essential component of the elongasome, its specific function remains unclear (*Sjodt et al., 2020*).

Based on biochemical and interaction studies and the confidence of the binary complexes, we modeled the elongasome incorporating MreBCD and MrdAB (*Figure 8*; *Graham et al., 2021*). Several studies have revealed connections between MrdC and MreD, MrdA and MrdB, and MreB and MreC, emphasizing the central role of MreB (*Graham et al., 2021*; *Liu et al., 2020*; *Banzhaf et al., 2012*), which forms filament-like oligomers in the cytoplasmic leaflet of the IM and recruits elongasome proteins (*Pichoff et al., 2019*). The predicted model of the elongasome suggests direct interactions between the MreB filament and the TM domains of MrdAB, but not with the other accessory proteins (*Figure 8*, *Figure 8—figure supplement 3*). Additionally, the model incorporates the MreCD-RodZ sub-complex, which is crucial for maintaining bacterial morphology. The cytoplasmic N-terminal domain of RodZ, characterized by a helix-turn-helix motif, likely contributes to protein-protein interactions with MreB, while the C-terminal domain may interact with periplasmic proteins to regulate bacterial morphology. The two sub-complexes are expected to interact with each other through the TM domains, likely facilitated by MrdB and MreD, as well as through the periplasmic domains of MrdA and MreC (*Figure 8—figure supplement 3*). These findings suggest that the cytoplasmic regions of MreB initially recruit the MrdAB GTase-TPase sub-complex, followed by the binding of MreCD-RodZ

to MrdAB. Interestingly, the overall arrangement of the elongasome model exhibits similarities to the divisome sub-complex FtsQLBWI. For instance, the connections between the periplasmic domains of MreC and MrdB in the elongasome resemble the interactions between FtsB and FtsI in the divisome. Additionally, the binding between the TM domains of MreCD and MrdA may serve a comparable role to the interactions of FtsQLB and FtsW in the divisome.

## Complexes involved in DNA replication

DNA replication involves the duplication of DNA during cell division to pass it on to the next generation. This intricate process is divided into three steps: initiation, elongation, and termination, which are carried out by conserved and dynamic protein machineries called replisomes. Despite progress made in characterizing the architecture of prokaryotic replisomes, the highly dynamic nature of replication makes the structural characterization challenging (*van der Ploeg et al., 2013*; *Reyes-Lamothe et al., 2010*).

The initiator protein of replication, DnaA, self-oligomerizes in the presence of ATP at the replication origin (OriC) (*Xu and Dixon, 2018*). This facilitates the formation of a DNA bubble, enabling the loading of helicases and recruitment of the DNA polymerase III complex (*Reyes-Lamothe et al., 2010*). First, the DnaBC complex, comprising 12 subunits, inhibits the unwinding of the double-stranded DNA. The later binding of DnaG primase to DnaB promotes dissociation from DnaC, resulting in DNA unwinding (*Reyes-Lamothe et al., 2010*). Experimentally solved structures of the DnaBC complex are available (6KZA), but data on oligomeric DnaA or DnaBG interactions is limited, as they can vary depending on bacterial species, cell cycle stage, and ATP/ADP presence (*Reyes-Lamothe et al., 2010*; *Xu and Dixon, 2018*). Previous studies have suggested that high concentrations of ATP-DnaA are required to adopt a helical filament-like structure to fully engage *oriC*. In our AF2 model, which describes tetrameric DnaA, the monomers are arranged in a bent filament, with the domain III of the monomers interacting in a head-to-tail manner and the domain IV facing the DNA (*Figure 9—figure supplement 1*; *Xu and Dixon, 2018*; *Katayama et al., 2017*). Unfortunately, we were unable to obtain larger oligomers or highly reliable interactions involving DnaG bound to DnaBC. One possible explanation for this is that the presence of a DNA molecule or accessory proteins, such as DiaA, are required in such cases.

DNA elongation is facilitated by the DNA polymerase III holoenzyme, which is a complex composed of three sub-complexes: the $\alpha\epsilon\theta$ polymerase core, the $\beta_2$ sliding clamp, and the $\delta \tau \eta \gamma 3\text{-} \eta \delta' \phi \chi$ clamp loader (*Reyes-Lamothe and Sherratt, 2019*). Detailed structural insights into these subassemblies have been obtained through cryo-EM studies, shedding light on their underlying mechanisms. However, modeling these large and dynamic complexes is challenging, especially in the absence of DNA molecules. Despite these inherent limitations, we identified an intriguing unresolved complex involving the interaction between the sliding clamp DnaN and DNA polymerase I (*Figure 9a*). The existence of this interaction suggests that DnaN may serve as a recruiter for DNA polymerase I at the replication fork, facilitating its attachment to the DNA. This finding highlights the crucial role of DnaN in coordinating the activities of multiple polymerases at the replication fork, thereby ensuring the efficiency and accuracy of DNA synthesis (*Reyes-Lamothe et al., 2010*).

During DNA replication, gyrases and topoisomerases IV form heterotetramers (GyrA$_2$B$_2$, ParC$_2$E$_2$) that modulate DNA topology by transiently cutting one or both DNA strands (*Fijalkowska et al., 2012*; *Badshah and Ullah, 2018*). Interestingly, we have discovered a potential connection between type II topoisomerases and the folate metabolism, facilitated by the GyrA-FolP interaction. As illustrated in *Figure 9b*, FolP and the C-terminal domain of GyrB share a similar interface with GyrA, indicating that FolP might compete with GyrB, thus exerting regulatory control over the complex. By exploring different stoichiometries, we have developed a model that suggests a complex comprising two GyrA and four FolP copies. When aligning our model with the FolP crystal structure bound to its substrate (1AJ0; *Figure 9b*, bottom), we observed a significant difference in the loop region spanning residues 22–36. In our model, this loop obstructs the catalytic site, whereas in the experimentally resolved structure, the pocket is accessible. This rearrangement of the loop, likely induced by the presence of the substrate, may be crucial in facilitating its interaction with GyrA while impeding its interaction with GyrB. Although the exact nature and significance of the interplay between these complexes remain incompletely understood, it is conceivable that this interaction plays a role in

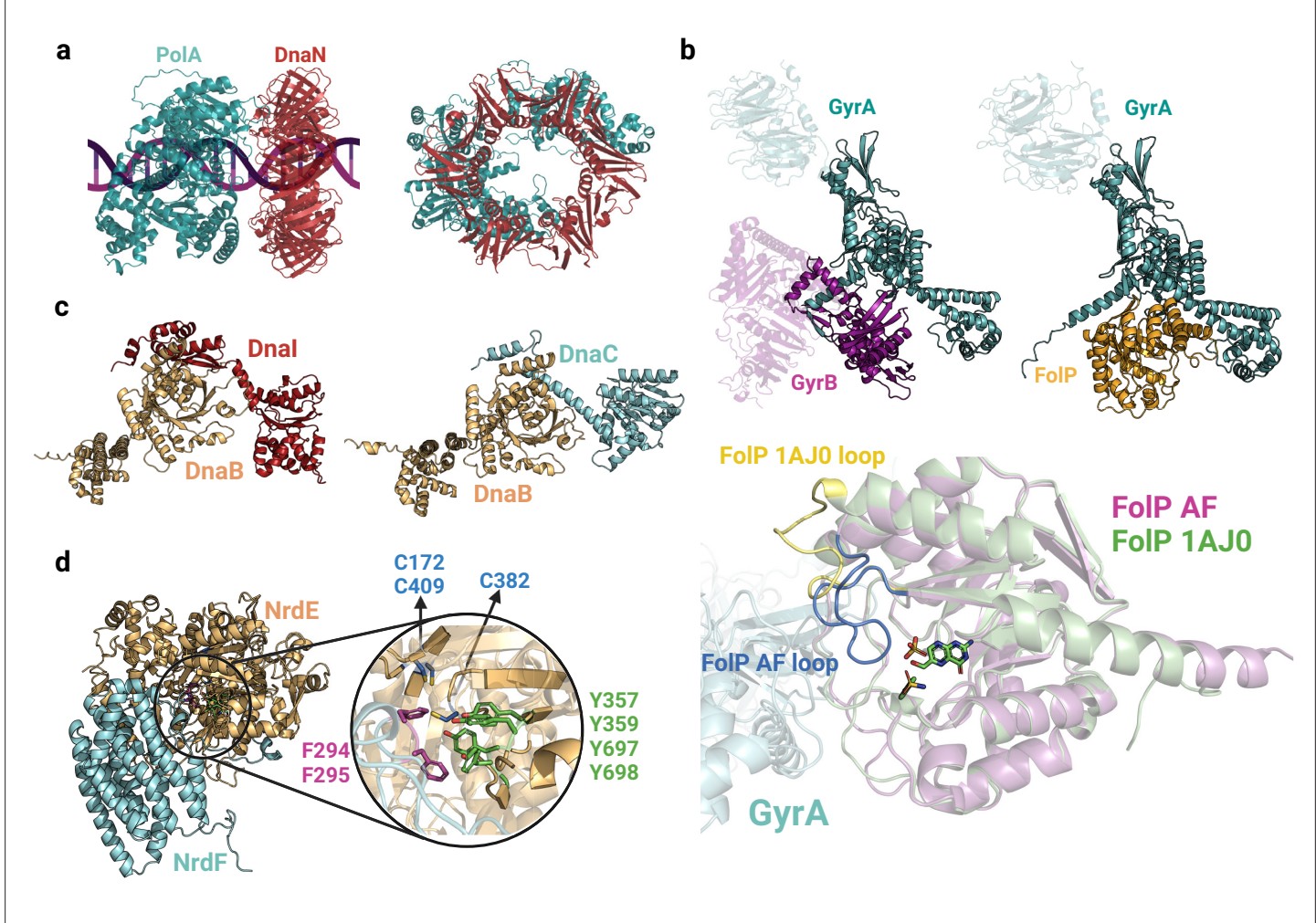

**Figure 9.** Complexes involved in DNA replication and synthesis. (**a**) Predicted interface between DNA polymerase I (PolA) and DnaN₂. (**b**) Models of GyrAB and GyrA-FolP (top). Close-up view of the GyrA-FolP interface and comparison with the crystal structure of FolP (bottom;1AJ0). The notable difference between the two structures is the loop region spanning residues 22–36, indicated in yellow/blue. (**c**) Predicted binary complexes DnaBI and DnaBC. The DnaBC predicted model is aligned to the solved crystal structure 6KZA (*Figure 2—figure supplement 1*). (**d**) Close-up view of the AlphaFold2 (AF2) predicted interface between NrdE and NrdF, highlighting important aromatic residues and cysteines involved in nucleotide reduction. Uniprot codes used for AF2: DnaB (DnaBI): A0A062WMW9, DnaB (DnaBC): P0ACB0, DnaC: P0AEF0, DnaI: Q8CWP7, DnaN: P0A988, GyrA: P0AES4, GyrB: P0AES6, FolP: P0AC13, NrdE: A0A0B7LYQ0, NrdF: A0A062WM39.

The online version of this article includes the following figure supplement(s) for figure 9:

**Figure supplement 1.** AlphaFold2 (AF2) prediction for DnaA₄ complex.

regulating DNA topology and preserving genome stability, given the vital role of folate metabolism in nucleotide synthesis.

Our Gram-positive interactome analysis reveals significant representation of both topoisomerases and replisome proteins. Notably, we have identified a distinctive interaction specific to Gram-positive bacteria involving the replication initiator DnaB and DnaI in *Bacillus subtilis* and *Streptococcus pneumoniae*. This PPI is absent in Gram-negative bacteria, as they lack a DnaI homolog and follow a different mechanism for replication initiation regulation (*Hooper and Jacoby, 2016*). In certain Gram-positive bacteria, DnaI interacts with DnaB, thereby aiding in the coordination of DNA replication initiation with the activities of the replication machinery. The predicted interface reveals close contacts between the N-terminal region of DnaI and the C-terminal domain of DnaB, resembling the structure of DnaBC (*Figure 9c*). Furthermore, our analysis predicts highly reliable binary interactions involved in DNA synthesis (nrdEF) and DNA transcription (rpoCZ, rpoC-greA, and rpoC-sigA). While the subunits of the DNA-dependent RNA polymerase have been extensively characterized, with cryo-EM structures

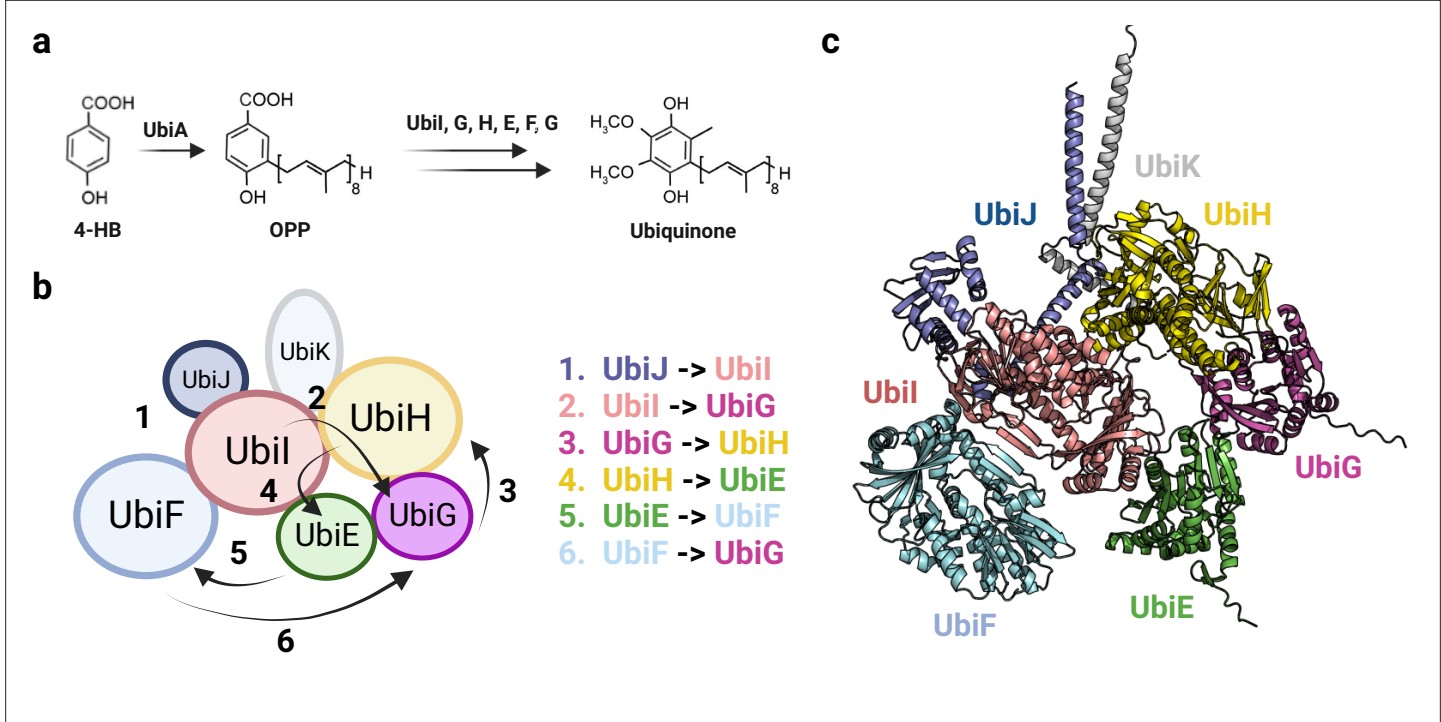

**Figure 10.** Organization of the Ubi metabolon. (**a**) Simplified ubiquinone synthesis pathway from 4-HB. 4-HB: 4-hydroxybenzoic acid, OPP: octaprenyl diphosphate. (**b**) Architecture of the Ubi metabolon. The numbers indicate the six reactions carried out by the Ubi metabolon, and the arrows depict the path followed by the lipid intermediate transported by UbiJ. In the first step, UbiJ shields the lipid intermediate and binds to UbiI, catalyzing the first reaction. In the following steps, the flexible UbiJ transport the biosynthetic intermediates to the next enzyme. (**c**) AlphaFold2 (AF2) model of the Ubi metabolon. Uniprot codes used for AF2: ubiA: P0AGK1, ubiE: P0A887, ubiF: P75728, ubiG: P17993, ubiH: P25534, ubiI: P25535, ubiJ: P0ADP7, ubiK: Q46868.

available at good resolutions, a high-resolution binary complex of the two components of the ribonucleotide reductase enzyme (NrdEF) remains unresolved. The predicted interface emphasizes the importance of the C-terminal loop of NrdF in the interaction, where the "thumb motif" containing two phenylalanine residues interacts with four tyrosines in the catalytic site of NrdE, probably to stabilize the nucleotide substrate (*Figure 9d*). These findings align with previous studies proposing that a thiyl radical is formed in Cys382 and the reduction of the nucleotide occurs through the cooperation of two cysteines present in the catalytic pocket, namely Cys172 and Cys409. These cysteines function as reducing agents (*Jameson and Wilkinson, 2017*).

## Complexes involved in the synthesis of ubiquinone

Ubiquinone, also known as coenzyme Q, plays a vital role in the electron transport chain, driving ATP synthesis in numerous organisms. In *E. coli*, a series of enzymatic steps performed by ubiquitin proteins (Ubi) utilizes chorismate and octaprenyl diphosphate as precursors to synthesize ubiquinone (*Figure 10a*; *Thomas et al., 2019*) While some Ubi proteins function independently, the final six reactions are performed by the Ubi metabolon (UbiE-I). This metabolon comprises three hydroxylases (UbiI, UbiH, and UbiF) and two methyltransferases (UbiG and UbiE) (*Abby et al., 2020*). The overall structure of this obligatory Ubi metabolon remains poorly defined. The metabolon enhances catalytic efficiency by organizing sequential enzymes of the same metabolic pathway and encapsulating reactive ubiquinone intermediates, thereby protecting against oxidative damage (*Abby et al., 2020*). Additionally, two accessory factors, UbiJ and UbiK, are present. UbiJ binds ubiquinone and other nonspecific lipids. The mechanisms by which octaprenylphenol exits the membrane and attaches to UbiJ in the soluble Ubi complex (potentially facilitated by UbiB) and how the final product is transported to the membrane are still unclear.

Through our analysis, we have identified high-confidence binary complexes involved in consecutive enzymatic steps, supporting the existence of the Ubi metabolon complex. Furthermore, we

have predicted the UbiE-K assembly, shedding light on the structural arrangement of this previously unexplored metabolon. Based on the predicted interfaces, UbiE and UbiH interact with UbiG and UbiI to form a heterotetramer. In addition, UbiF seems to interact only with UbiI (*Figure 10b and c*). Additionally, the accessory proteins UbiJ and UbiK adopt a coiled-coil structure, which suggests their association with the membrane to facilitate the delivery of ubiquinone. Moreover, the SCP2 domain of UbiJ creates a lipophilic environment that accommodates lipid intermediates within the Ubi complex, consistent with previous findings (*Hajj Chehade et al., 2019*). Our model further suggests that the presence of two α-hairpin domains in UbiJ facilitates its interaction with UbiK, with the loops assisting the movement of the SCP2 domain between different subunits. The initial reaction catalyzed by the metabolon is likely initiated by the interaction between UbiJ and UbiI (*Abby et al., 2020*; *Hajj Chehade et al., 2019*). Subsequently, the lipid intermediate is sequentially transported to UbiG, UbiH, UbiE, UbiF, and ultimately to UbiG to catalyze the final reaction (*Figure 10b and c*). Interestingly, the initial reaction involves a hydroxylase, succeeded by a methyltransferase, and this process is reiterated once, ultimately concluding with another hydroxylase. Additionally, the three hydroxylases share a very similar structure, and likewise, the two methyltransferases also display structural homology. It should be noted that the quaternary structure of our model suggests the possibility of Ubi subunit polymerization, as it deviates significantly from the 1 MDa Ubi metabolon suggested by *Abby et al., 2020*. This initial model of the complete Ubi metabolon provides valuable insights into the complex's mechanism, emphasizing the role of UbiJ in transporting lipid intermediates between different subunits.

## Conclusion

The advancements in deep-learning technologies are poised to revolutionize various life science fields, particularly structural bioinformatics. Developing comprehensive interactomes holds great promise in identifying potential targets for the discovery of novel antibiotics. By combining deep-learning model confidence scores with interactome data, we can address the issue of high false positive rates. The structural insights presented in this study shed light on the underlying mechanisms of crucial biological processes in prokaryotes. Many of the discussed complexes lacked prior structural characterization, making the findings valuable for structural-based drug discovery approaches. To further enrich our interactomes, we can incorporate protein interaction data from other species or include information about the quaternary structure of the complexes. We hope that with the continuous training of deep-learning models using larger datasets, we will generate more accurate and confident protein complex models in the near future.

It is also crucial to acknowledge the limitations of the methodology employed in this study. First, the interpretation of protein essentiality can be influenced by the culturing conditions of bacteria. The essential proteins mentioned in the literature have been identified in bacteria cultured under rich medium conditions. However, it is important to recognize that protein complexes are dynamic entities that can rearrange in response to changing conditions and cellular stress. Therefore, it is necessary to understand these interactions within the appropriate biological context. Second, studying isolated binary complexes may result in inaccurate representations of the complete architecture due to the absence of accessory proteins or the omission of the correct stoichiometry. Finally, the performance of the AF-Multimer algorithm tends to decrease with a higher number of chains and in the case of heteromeric complexes. This is because homomeric structures typically possess internal symmetry, resulting in identical interfaces between chains and consistent interface quality. Heteromeric complexes, on the other hand, are more susceptible to variations in confidence scores due to irregularities in interface regions. Despite these constrains, AF2 showed remarkable predictive accuracy in modeling bacterial protein-protein complexes, generating high-confidence models for almost 90% of the complexes tested. Nevertheless, our results present an initial description of the essential interactome, which can assist researchers in gaining a deeper understanding of the fundamental processes within bacterial cells. As additional data becomes available in the coming years and new methods are developed to enhance the accuracy of protein multimer prediction, structural biology will deeply improve our understanding of the cell interactome.

## Methods

### Compilation of essential proteins and processing the data

First, we compiled from previous studies the essential proteins for four Gram-negative (*Acinetobacter baumannii*, *Bai et al., 2021*), *E. coli* (*Baba et al., 2006*; *Gerdes et al., 2003*; *Goodall, 2018*), *Klebsiella pneumonia* (*Ramage et al., 2017*), and *Pseudomonas aeruginosa* (*Liberati et al., 2006*; *Gallagher et al., 2011*; *Poulsen et al., 2019*) and four Gram-positive species (*Bacillus subtilis*, *Commichau et al., 2013*). *Clostridium difficile* (*Dembek et al., 2015*). *Staphylococcus aureus* (*Ji et al., 2001*; *Chaudhuri et al., 2009*; *Santiago et al., 2015*), and *S. pneumoniae* (*Liu et al., 2017*; *Source data 1*; *Figure 1—figure supplements 6–7*). In addition, we retrieved all synthetically lethal interactions found in *E. coli-K12-BW25113* from the Mlsar database (*Zhu et al., 2023a*). Then, we mapped the Uniprot ID, the locus tag, and the gene name for each essential protein using Uniprot ID mapper to maintain the same annotation for all the entries and accommodate our comparisons in future mapping steps (*Source data 1*). We used EGGNOG mapper v2 (*Launay et al., 2022*) to retrieve the ortholog proteins of all our compiled proteins. By mapping the ortholog proteins we could link the proteins belonging to different species.

To retrieve the essential PPIs, we used the 'Multiple protein' search from the STRING database v11.0 (*Szklarczyk et al., 2021*) website (https://version-11-0.string-db.org). We selected those interactions with a high-confidence score (combined score >0.7) and/or those based purely on experimental data (experimental score >0.15) then we downloaded the short version of the output containing only one-way edges. The networks downloaded from STRING can also include interactions involving non-essential proteins, which we filtered out. In addition, to increase the confidence of the selected essential interactions, we shortlisted the Gram-negative/Gram-positive PPIs identified in at least two out of the four species. Finally, ribosomal-related proteins and tRNA ligases were also discarded, because they form huge multiprotein complexes and/or they are proteins too massive to be predicted by AF2 in our setup. A total of 722 Gram-negative and 680 Gram-positive essential PPIs were modeled. Furthermore, 722 Gram-negative and 680 Gram-positive random essential PPIs were generated to test whether AF2 can discriminate between high-accuracy and incorrect folds as well as to define an ipTM score cutoff. We verified that the randomly generated PPIs were absent in the positive dataset.

### Compilation of experimentally solved PPIs not included in the training dataset of AlphaFold 2.3.1

We compiled all bacterial protein complexes from the PDB (accessed on 2023-09-15) that were not included in the training set of AF v2.3 (complexes until 2021-09-30). Our selection criteria encompassed heterodimers released after 2021-09-30 that were determined by either X-ray crystallography or cryo-EM with a resolution of 2 Å or better. We then selected the polymer entities grouped by UniProt Accession, retrieving a total of 425 structures. To eliminate redundancy, we clustered these structures using the 'easy-cluster' utility from Foldseek, with an alignment coverage cutoff of 0.9. From these clusters, we selected only one representative structure for each cluster, resulting in 304 representative structures. Next, we used the 'easy-complexsearch' module from Foldseek to align these structures with the AF training set and retained only those structures with a sequence identity below 30% with complexes in the AF training set, ultimately obtaining a total of 140 low-homology structures. We calculated the TM-score with the TMalign package downloaded from https://zhanggroup.org/TM-align/. Additionally, the DockQ and iRMS scores were determined using the 'DockQ.py' script downloaded from https://github.com/bjornwallner/DockQ; (*Wallner, 2016*; *Basu and Wallner, 2016*).

### Prediction of binary protein complexes and interactomes

We used AlphaFold v2.3.1 (https://github.com/deepmind/alphafold; *Jumper et al., 2022*) to predict the structures of our essential PPIs. We installed locally AF2 in a cluster with the following node configuration: Intel(R) Xeon(R) Gold 6226R CPU @2.90 GHz and a NVIDIA GeForce RTX 3080 Ti GPU. The database versions used to carry out the predictions are the following: UniRef90 v2022_01, MGnify v2022_05, Uniclust30 v2021_03, BFD (the only version available), PDB (downloaded on 2023-01-10) and PDB70 (downloaded on 2023-01-10). The FASTA files containing the sequences of the essential proteins were fetched from Uniprot. To run AF-Multimer we executed the Python script 'run_alphafold.py' pointing to the FASTA files and adding the 'model_preset = multimer' flag. We

retrieved the model with the best ipTM score over the five predicted models, which are stored in the 'ranking_debug.json' file, and computed pDockQ and pDockQ2 scores for the selected models (*Akdel et al., 2022*; *Bryant et al., 2022b*). The PPIs and the scores were collected in tabular format (*Source data 1*) and introduced to Cytoscape to build the essential interactomes (*Figure 2*). One protein partner was defined as 'Source node' and the other one as 'Target node' to establish the interactions (undirected edges) between the proteins (nodes). The ipTM score was expressed as 'Edge attribute' to modify the colors and widths of the edges depending on the ipTM score values. When possible, models were compared with available experimental structures deposited in the PDB.

### Protein interface and surface analysis

We analyzed the interfaces with the 'GetInterfaces.py' Python script from the Oxford Protein Informatics Group (OPIP, *Krawczyk, 2013*) to obtain interacting and interface residues. The contact distance was defined as 4.5 Å and the interface distance as 10 Å. To find the surface residues we employed the findSurfaceAtoms PyMol function with a cutoff of 6.5 Å (*de Groot et al., 2020*). Per-residue conservation scores were computed using VESPA (*Cantalapiedra et al., 2021*), whose scores range from 1 (most variable) to 9 (most conserved). SASA was computed using FreeSASA (*Mitternacht, 2016*). Python module. Statistical data analyses were carried out using R v4.2.1 and Python v3.9. Molecular graphics were performed with PyMol.

## Acknowledgements

This study was funded by a Research Grant 2022 of the European Society of Clinical Microbiology and Infectious Diseases (ESCMID) and the Spanish Ministerio de Ciencia e Innovación (PDC2021-121544-I00 funded by MCIN/AEI/10.13039/501100011033 and European Union Next GenerationEU/PRTR, and project PID2020-114627RB-I00 funded by MCIN/AEI /10.13039/501100011033), all to MT. This work has been co-financed by the Spanish Ministry of Science and Innovation with funds from the European Union NextGenerationEU, from the Recovery, Transformation and Resilience Plan (PRTR-C17.I1) and from the Autonomous Community of Catalonia within the framework of the Biotechnology Plan Applied to Health. JGB is a recipient of a Joan Oró Fellowship from the Generalitat de Catalunya (2023 FI-100278). We would like to thank Dr. Enea Sancho Vaello for her comments on this work.

## Additional information

### Funding

| Funder | Grant reference number | Author |
| --- | --- | --- |
| Ministerio de Ciencia e Innovación | PDC2021-121544-I00 | Marc Torrent Burgas |
| European Society of Clinical Microbiology and Infectious Diseases | ESCMID2022 | Marc Torrent Burgas |
| Ministerio de Ciencia e Innovación | PID2020-114627RB-I00 | Marc Torrent Burgas |
| Generalitat de Catalunya | Joan Oró Fellowship 2023 FI-100278 | Jordi Gómez Borrego |

The funders had no role in study design, data collection and interpretation, or the decision to submit the work for publication.

### Author contributions

Jordi Gómez Borrego, Data curation, Formal analysis, Investigation, Methodology, Writing – original draft; Marc Torrent Burgas, Conceptualization, Resources, Supervision, Funding acquisition, Writing – original draft, Project administration, Writing – review and editing

## Author ORCIDs

Jordi Gómez Borrego (ID) http://orcid.org/0000-0003-1963-5455
Marc Torrent Burgas (ID) https://orcid.org/0000-0001-6567-3474

## Decision letter and Author response

Decision letter https://doi.org/10.7554/eLife.94919.sa1
Author response https://doi.org/10.7554/eLife.94919.sa2

## Additional files

### Supplementary files

• Supplementary file 1. List of validated bacterial complexes. The listed complexes were not included in the training dataset of AF and share <30% sequence identity with all models deposited in the PDB.

• MDAR checklist

• Source data 1. Essential protein annotations and protein-protein interactions (PPIs) scores provided by AlphaFold2 (AF2).

### Data availability

All models described in this paper are available on ModelArchive (https://modelarchive.org, dataset ID: ma-sysbio-bei) with accession codes in Table 1. The scores of selected and random binary PPIs and the annotations of the essential proteins are provided in Source data 1.

The following dataset was generated:

| Author(s) | Year | Dataset title | Dataset URL | Database and Identifier |
|---|---|---|---|---|
| Gómez Borrego J, Torrent Burgas M | 2024 | Structural assembly of the bacterial essential interactome | https://doi.org/10.5452/ma-sysbio-bei | ModelArchive, 10.5452/ma-sysbio-bei |

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
