## [Editor Report]

This important study uses AlphaFold2 to predict the structures of bacterial protein complexes that the authors classify as "essential". The evidence supporting the conclusions is convincing, as the authors have tested the approach on an external dataset of 140 experimentally solved bacterial protein-protein complexes, 81% of which were predicted with high accuracy. This paper will be of general interest to a wide audience in the field of biosciences and in particular for molecular biologists.

---

## [Decision Letter]

[Editors' note: this paper was reviewed by Review Commons.]

---

## [Author Response]

Reviewer #1The paper provides models of essential complexes formed in bacteria. These models have been predicted by AlphaFold2 and in some of the models, information from existing experimental structures is utilized. The predicted models have been calculated based on standard workflow procedures which are explained in detail and can be reproduced by others. The figures are informative and clear.

We are grateful for the reviewer's insightful comments, which have significantly contributed to improve our manuscript.

Suggestions for improvement:a. The PDB accession codes of the experimental structures should be providedb. A comparison of the predicted models with the experimental structures should be provided (e.g., same orientation, superposition). In Figure 6 for example, a figure with superposition or use of the same orientation would be more informative.

As suggested by the reviewer, we have included a new table (Table 1) listing all experimental structures discussed in the main text, with the corresponding PDB codes. All predictions are listed in Source data 1. For instances with available PDB codes, we compared the predicted structures to the experimental ones (new Figure 2—figure supplement 1). In Figure 6, the structures were difficult to superimpose because the subunits in the complexes have different relative orientations. To help comparing both models, we have added a schematic representation (new Figure 6c).

The paper will certainly generate many hypotheses based on the predicted models. In this respect, it would be useful for a wide audience in the bioscience field. However, the discussed models will need experimental verification by various techniques, such as X-ray crystallography, cryo-EM, SAXS, and structural proteomics. A more thorough analysis of the literature may help to improve the paper in this respect.

We acknowledge that we may have under-emphasized this aspect in our manuscript and understand the need of validating the predictions to determine the general validity of our AlphaFold (AF2) models. To address this issue, we conducted a thorough validation of 140 bacterial protein-protein complexes from the PDB. This dataset contains all structures published after the last release of AF2 that share less than 30% sequence homology with all other complexes deposited in the PDB. Hence, this dataset serves as the strongest test possible for AF2's prediction accuracy in bacterial complexes. We found that 81% (113 out of 140) of these structures were accurately predicted by AF2 according to our criterion (ipTM>0.6). From all models generated, 83% (116 out of 140) were almost identical to the actual structures in terms of correct folding (TM-score > 0.8). Most interestingly, 72% (101 out of 140) of the predicted structures were acceptable in terms of root mean square deviation at the interaction interface (i-RMSD < 4 Å) and 56% (79 out of 140) were virtually identical to the real structures (i-RMSD < 2Å), highlighting the excellent prediction power of AF2. This high precision likely reflects the vast number of bacterial sequences available in databases, from which coevolutionary relationships among protein residues can be inferred with great accuracy. These results provide strong support for our findings. We have included all this new data in a revised version of our manuscript, both in the main text (Results and Discussion, page 3) and Supplementary file 1, that contains all the results.

Also, we investigated whether our complexes have known crosslinks in the xlinkdb database (https://xlinkdb.gs.washington.edu/xlinkdb/). We also revisited the literature for more datasets that include crosslinking data and found an additional study that was potentially useful to validate our results (Lenz et al., Nat. Comm., 2021). Overall, we found information for 14 bacterial complexes. In all cases but one, the distance restraints identified (crosslinked lysines are ~15-20Å apart) are compatible with our models. Hence, although the overlap between the datasets and our list of validated interactions was not extensive, for the complexes that did match, our models were consistent with the experimental data. We have incorporated these results in the main text (Results and Discussion, page 3) and Source data 1, that contains all the results.

Finally, we identified several mutations in BirA, documented in the literature, that affect its interaction with AccB. In BirA mutations M310L and P143T were found to induce a superrepressor phenotype (BirA lacks the capacity to biotinylate AccB). These mutations do not significantly affect the BirA active site but can destabilize the BirA-AccB interface (Chakravarti et al., J. Baceriol., 2012), supporting our results. We have added this information in the main text (page 8).

Reviewer #2This study attempts to identify the 'essential interactome' through combining information in presence/absence genomics across bacteria, information in the STRING database, and predictions from α-fold. Overall, the strategy is clear, and I do not have concerns about reproducibility and clarity.

We value the reviewer's constructive evaluation of our manuscript, and we would like to thank the reviewer's feedback as it has significantly helped us in improving our manuscript.

Strengths: Clever approach to get at the essential interactome.Weaknesses: Putative impact. It is clear why understanding which interactions are present are important. But even as the authors suggest, interactions are dynamic and there are plenty of other tools that people could use to find interactions (including AA Coev that the authors themselves cite). The counter argument the authors bring up is the high false positive rate of interactions that is solved by this method. While true, the stringency criteria for what constitutes an interaction in this paper is remarkably high: each protein within the interaction needs to be essential, and needs to have a high confidence score in STRING, and then there is a hyperparameter that dictates the level at which AlphaFold 2 is providing confident answers. In this sense, this is less about an 'essential' interactome, and more about an interactome that is present with the highest true positive rate (trading off with the ability to discover new interactions at a reasonable breadth).

We appreciate the reviewer's insights concerning the stringency criteria for defining interactions. Here, we provide a detailed justification for our selection criteria and show how it aligns with our goal of identifying high-confidence interactions.Protein essentiality: In our model, interactions are considered essential if, and only if, both proteins involved are essential, providing a conservative estimate for the essential interactome. In our revised manuscript, we explored the possibility for two non-essential proteins to form an essential interaction by investigating synthetically lethal interactions. Out of all synthetic lethal interactions in *E. coli*, only 28 interactions were identified, and only two could be modeled with an ipTM score > 0.6. Likely, these non-essential proteins operate in parallel or compensatory pathways instead of interacting directly. These findings lend support to our hypothesis and suggest that our interactome encompasses most essential interactions. These results were included in the revised manuscript.

**Author response image 1. sa2fig1:** 

**Author response table 1. sa2table1:** Performance of state-of-the-art PPI prediction methods (Huang et al., 2023).

Methods	AUPRC^[Table-fn appR3table1fn1]^
SGPPI	0.422
Profppikernel	0.359
PIPR	0.342
PIPE2	0.220
SigProd	0.264

*AUPRC denotes the average AUPRC value of 10-fold cross-validation.

It is clear from the data that such methods are not mature enough to be used as confident predictors. Hence, we decided to resort to validated interactions in the String database, which is one of the most comprehensive PPI databases. In this revised version, we have expanded our data set to include all experimentally abelled interactions in the String database, even those with a low probability (experimental score > 0.15). The addition of these new interactions increased the total number of interactions tested from 1089 to 1402 and generated 38 new models for Gram-negative species (13 with high accuracy) and 275 new models for Gram-positive bacteria (18 with high accuracy). All interactions are now included in the Source data 1 and high accuracy models will be deposited on Model Archive after acceptance.

Alphafold (AF2) criterion for complex prediction. Although AF2 has its limitations, its accuracy in predicting bacterial complexes is consistently high. Additionally, as extensively detailed in the response to Reviewer 1, we conducted a thorough validation of 140 bacterial protein-protein complexes from the PDB, whose structure was published after the last release of AF2. Our results show that AF2 was able to predict 81% of these complexes, with high accuracy, in line with other reported studies (Evans et al., Yin et al., 2022). While there might be some minor deviations, AF2 can largely capture the bacterial essential interactome accurately. In the revised version, we compare pDockQ and pDockQ2 metrics with our ipTM criterion to define confident models. We observed that both pDockQ and pDockQ2 metrics were capable of identifying highly reliable complexes, but also disregarded actual complexes (Figure 1—figure supplements 1-2). Thus, we decided to retain our initial criterion, based on ipTM scores, which is consistent with other authors who used similar ipTM thresholds to model bacterial interactions (e.g., O’Reilly et al., 2023).

In summary, although our methodology has inherent limitations, we believe that our approach is sound and can give a comprehensive and realistic view of the bacterial essential interactome. We hope that these new insights further substantiate our approach.

I don’t know of too many studies that use AlphaFold 2 in this way. This was clever. However, there are plenty of studies that use phylogenomic information to infer interactions. In this sense, the core idea of the paper is not intrinsically novel.

We thank the reviewer for valuing our approach. Although other methods have been used to predict interactomes, our study, to the best of our knowledge, provides the first high-quality essential interactome for bacteria. We used experimental data (analysis of single deletion mutants) to define the essential interactions in bacteria. Other methods, either using phylogenomic information and/or deep learning tools to infer interactions, have a poor performance, as illustrated in the preceding table. Often, these methods yield a high number of interactions and, in many cases, show a bias towards overrepresented entries in the positive databases used to train the predictors (Macho Rendón et al., 2022). Also, while other methods lack detailed structural insights into the interactions, we offer structural models for every interaction tested.

Overall, I do feel this would be worth publishing as an expose of AF2 is capable of. I'm not sure of the impact it will have on researchers, however.

We appreciate the reviewer's positive feedback on our manuscript. Using AF2, we identified key interactions using only gene deletion mutant data. This manuscript reveals new insights into the assembly of essential bacterial complexes, providing specific structural details to understand their stability and function. Additionally, our work seeks to establish a methodology applicable to all bacterial species, guiding future research in this field. The approach taken in this study may expand drug targeting opportunities and accelerate the development of more effective antibiotics aimed to disrupt these essential interactions. In conclusion, the impact of the paper lies in its novel use of Alphafold2 to understand essential bacterial protein interactions, providing key insights into assembly mechanisms, and identifying new potential drug targets.

Response to Reviewer #3The selection of "essential" interactions is a bit arbitrary, given that their main criterion for selection is that both proteins are essential. Unfortunately, it's not always clear where the essential protein data is coming from. Authors cite Mateus et al. (ref 15) as source for *E. coli*, but I don't see an explicit list of essential genes in this paper (nor its supplement). For Pseudomonas the citation doesn't contain author information and for Acinetobacter essentiality only seems to refer to "essentiality" in the lung.As a minimum, the author should provide a table with summary statistics for the essential proteins they are using, as this is the basis for the whole paper. Such a table should include the names of the species, the number of genes that are considered as essential, a very brief characterization of how essentiality was determined and the source for this information. For instance, are the genes listed in the Supplementary File congruent with the genes in the Database of Essential Genes (DEG) for these organisms? Finally, authors should indicate in that table which (essential) protein pairs are conserved across species, as this is another one of their selection criteria. Conservation is not necessary for an essential interaction, but it certainly makes it more likely.

We understand the reviewer's concerns regarding the selection of essential interactions and the need for a more thorough description of the sources of essential protein data. To address these concerns in the revised manuscript:

We included a clear explanation of the sources for essential protein data, including proper citations for each organism in Source data 1. The selected studies were primarily sourced from the DEG database. If data was unavailable, we revised the literature for relevant studies. The DEG database's most recent update was on September 1, 2020. A graphical summary of the datasets has been included in Figure 1—figure supplements 6-7, that shows the overlapping between the different studies.

Also, the reviewer was right in pointing out that for *Acinetobacter baumannii*, the study was conducted in the lung, which may bias the results as all other studies were performed in the test tube. To solve this, we replaced this study for Bai et al., 2021, that was performed in rich medium.

Author should also state whether they have verified that none of the random pairs are in the positive set.

We thank the reviewer for this comment. We certainly checked that none of the random pairs was present in the positive dataset. This clarification has now been added to the methods section.

This is also relevant because authors "retrieved all high-confidence PPIs between these proteins from the STRING database" which provides compound scores for interactions but that has often little to do with physical interactions (given that the scores factor in co-expression and several other criteria). In fact, I find STRING scores difficult to interpret for that very reason.

We appreciate the reviewer's comment to the use of combined interaction scores from the STRING database. We agree with the reviewer that STRING combined scores are somehow difficult to interpret because they combine different evidence of interaction. We decided to use the STRING combined scores to include interactions that may not have direct experimental evidence but are probable to interact according to other information (e.g., co-expression). However, to further examine the interactome we have also included in the revised version all interactions with experimental evidence in String to complete our interactome. As mentioned in the response to Reviewer 1, we expanded the tested interactions from 1089 to 1402. This resulted in 38 new models for Gram-negative species, with 13 being highly accurate, and 275 for Gram-positive bacteria, of which 18 were highly accurate. All interactions are now included in the Source data 1 and high accuracy models will be deposited on the Model Archive after acceptance.

The authors "reasoned that a given interaction would only be essential if and only if both proteins forming the complex are essential" – this sounds reasonable but doesn't capture synthetically lethal (genetic) interactions, that is, interactions between two proteins that are both non-essential but are essential in combination. Admittedly, I don't have a number of how many such cases exist, but there are such cases in the literature (e.g. Hannum et al. 2009, PLoS Genet 5[12]: e1000782, for yeast).

We thank the reviewer for bringing this point into discussion. We acknowledge that our reasoning does not capture synthetic lethality, which occurs when the loss of one of two individual genes has no effect on cell survival, but the simultaneous loss of both leads to cell death. In this case, the two genes or proteins are non-essential individually but become essential in combination. To cover synthetic lethality, we retrieved all synthetically lethal interactions found *in Escherichia coli*, strain K12-BW25113 from the Mlsar database and included them in our pipeline. We identified 28 synthetically lethal PPIs (involving 45 proteins) and we modeled them with AF2. Only two interactions displayed an ipTM score > 0.6 (nadA-pncB and nuoG-purA). Hence, the number of interactions due to synthetic lethality seems to contribute low to the overall interactome. We believe that synthetic lethal partners often function in parallel or compensatory pathways, rather than directly interacting with each other. For example, in yeast, the genes RAD9 and RAD24 are synthetic lethal. RAD9 is involved in cell cycle checkpoints, while RAD24 is involved in DNA damage response. They function in related pathways but do not encode proteins that directly interact with each other. Hence, finding specific examples of proteins that are both synthetic lethal and directly interact might be challenging as the synthetic lethal relationship often reveals functional rather than physical interactions.

Apart from that, one could question the selection method more generally, given that for a biological process always essential and non-essential proteins work together, so I wonder why the authors didn't include additional proteins known to be involved in specific processes as this could make their predictions much more biologically meaningful.

We agree with the reviewer that accessory proteins are important to understand the biological context of interactions. In fact, in several sections of our manuscript, we included accessory proteins to fully describe the essential complexes. For example, in the cell division complex, we incorporated proteins like MreCD-RodZ from the elongasome to enhance the structural context of the interactions. However, a comprehensive explanation of all identified interactions and accessory proteins would extend beyond the scope of this manuscript and further lengthen an already extensive document. In our study, we sought to describe the fundamental interactions for both Gram-negative and Gram-positive bacteria. We anticipate that our findings will prompt additional research to confirm our hypotheses and enhance knowledge of these protein complexes within the proper cellular context.

In any case, to understand their choice better, authors should provide a table (in the main text) summarizing the proteins they actually analyze and discuss in more detail in their models. This would allow a reader to see which proteins are considered essential and which ones are missing. I would organize this by function / pathway / process, so these proteins are listed in a functional context.

We added Table 1 in the main text, listing all interactions described in the text. Table 1 includes the proteins involved in each complex, the ipTM score of the interaction, whether a PDB code is available for comparison and the functional classification of the interaction.

With regard to docking, please also discuss why you focus on iPTM, as there are other derived metrics from AF2 scores, such as pdockq based on if_plddt (e. g. Bryant et al., 2022), as well as external metrics to AF2 (physics-based methods such as Rosetta). Another option may be a modified versions of AF2 multimer, such as AFSample, which produces a greater diversity of models, allowing for more "shots on goal" and ultimately a higher success rate, assuming one has a reliable QC filter (I wonder how those compares to iPTM).

We did not use AFsample because is a very expensive computational approach that would require too many resources for the batch prediction of more than 2.000 complexes. Afsample generates 240x models, and including the extra recycles, the overall timing is around 1,000x more costly than the baseline. However, we acknowledge that using other metrics can be useful to further evaluate our models. Hence, we investigated how pDockQ and pDockQ2 metrics compare with ipTM score. We observed that pDockQ hardly correlates with ipTM (R = 0.328) whereas the improved metric pDockQ2 correlates much better (R = 0.649). All complexes described in the manuscript, which have an ipTM score higher than our threshold (0.6), have also a pDockQ2 score higher than 0.23, except for six interactions that have a lower pDockQ2 score. However, these scores improve when the interactions are modeled with accessory proteins in the complex. This somehow suggests that the IpTM metric better captures binary interactions when these are excluded from their context. It is possible however, that pDockQ scores are better in discriminating false positive interactions than ipTM scores. Based on the strong correlation between the two metrics and the observation that ipTM may better capture binary interactions, we decided to keep our method in the manuscript. Other authors have employed analogous ipTM thresholds to model bacterial interactions (e.g., O’Reilly et al., 2023). Notwithstanding, we also included pDockQ and pDockQ2 metrics in Source data 1, so readers can evaluate complexes based on these metrics.

P. 1, third last line: “the essential interactome is a potentially powerful strategy to […] identify new targets for discovering new antibiotics”Figures and figure legends need to be explicit which species is represented (ideally with a Uniprot ID) and which structure was predicted by alphafold and which one has an experimental structure. Known structures should be indicated in a table, as suggested above.Figure 5: LptF is too dark when printed, so a lighter color may be better.Figure 6: The cryoEM and alphafold structures look quite different, so please discuss discrepancies between them (in terms of prediction or cryEM odelling). A schematic may be helpful to illustrate the differences in more clarity.Figure 7: LolC is also too dark when printed. Make lighter.Maybe in some cases it may be worthwhile looking at Consurf structures to see if the predicted inferfaces are indeed more conserved than the non-conserved parts.

We thank the reviewer for his/her insightful feedback on our manuscript. We have addressed all these comments as follows:

The statement on page 1 was revised as suggested.

The main significance of this study is its potential use for a better understanding of the protein complexes described in more detail (and the fact that alphafold can be applied in a similar fashion to many other complexes). This is why the individual sections need to be evaluated to process-specific experts (disclaimer: I have only worked on some of the complexes, but I am not an expert on any of them). I wonder if it would make more sense to break out some of the sections on individual complexes into separate papers, and then discuss them in more detail and with more context from previous studies. Complexes such as the divisome have a huge body of literature and it may be worth reviewing which structures are known and which ones are not. However, the dynamic and labile nature of these complexes have made it difficult for both crystallography as well as odelling to get a good structural understanding, but some of the models proposed here may be useful for overcoming some of these hurdles.

We appreciate the reviewer's suggestion. While we acknowledge the complexity of some of the individual complexes, such as the divisome, and the wealth of existing literature, we believe that the current manuscript provides a valuable comprehensive view on how AF2 can be used to predict essential protein complexes in bacteria. In our opinion, dividing the manuscript in separate pieces might dilute its scope. Nonetheless, we are exploring in our laboratory the interactions detailed in the manuscript, aiming to further expand the knowledge on these important complexes and their potential as targets for new antimicrobials.

References:

Bai J, Dai Y, Farinha A, et al. Essential Gene Analysis in Acinetobacter baumannii by High-Density Transposon Mutagenesis and CRISPR Interference. J Bacteriol. 2021; 203(12):e0056520.

Chakravartty, V. and Cronan, J. E. Altered regulation of *Escherichia coli* biotin biosynthesis in BirA superrepressor mutant strains. J Bacteriol*.* 2012; 194:1113–1126.

Evans R, O’Neill M, Pritzel A, et al. Protein complex prediction with AlphaFold-Multimer.

bioRxiv. 2021; 2021.10.04.463034.

Huang Y, Wuchty S, Zhou Y, Zhang Z. SGPPI: structure-aware prediction of protein-protein interactions in rigorous conditions with graph convolutional network. Brief Bioinform. 2023; 24(2):bbad020

Lenz, S, Sinn, L.R, O’Reilly, F.J. et al. Reliable identification of protein-protein interactions by crosslinking mass spectrometry. Nat Commun. 2021; 12:3564.

Macho Rendón J, Rebollido-Ríos R, Torrent Burgas M. HPIPred: Host-pathogen interactome prediction with phenotypic scoring. Comput Struct Biotechnol J. 2022; 20:6534-6542.

O'Reilly FJ, Graziadei A, Forbrig C, et al. Protein complexes in cells by AI-assisted structural proteomics. Mol Syst Biol. 2023; 19(4):e11544.

Potvin, E., Lehoux, D.E., Kukavica-Ibrulj, I., et al. in vivo functional genomics of *Pseudomonas aeruginosa* for high-throughput screening of new virulence factors and antibacterial targets. Environmental Microbiology. 2003; 5: 1294-1308.

Wang N, Ozer EA, Mandel MJ, Hauser AR. Genome-wide identification of Acinetobacter baumannii genes necessary for persistence in the lung. mBio. 2014; 5(3):e01163-14.

Yin, R, Feng, BY, Varshney, A, Pierce, BG. Benchmarking AlphaFold for protein complex modeling reveals accuracy determinants. Protein Science. 2022; 31(8):e4379.